# NIRANTAR: CONTINUAL LEARNING WITH NEW LANGUAGES AND DOMAINS ON REAL-WORLD SPEECH DATA

## ABSTRACT

We present Nirantar[1] based on a large-scale effort to collect extempore and conversational speech data from participants spanning 22 languages across diverse locations in India. Given the extensive number of languages and locations involved, data is collected in incremental batches. Each batch introduces new languages, new domains (locations), or both, creating a practical playground for continual learning (CL). Nirantar contains a total of 3250 hours of human-transcribed speech data covering 208 Indian districts across 22 languages, with 1720 hours newly released as a part of this work. The data inflow and resulting multilingual multi-domain episodes are based on real-world data collection rather than simulated episodes commonly found in existing CL datasets. In particular, the amount of data collected and the number of languages and domains involved are not uniform across episodes, reflecting a practical and real-world continual learning scenario. This dataset serves as a playground for training and evaluating CL approaches in three different scenarios: Language-Incremental (LIL), Domain-Incremental (DIL), and the novel Language-Incremental Domain-Incremental Learning (LIDIL), which has not been studied before. To establish the dataset's usefulness, we evaluate several existing CL approaches within these scenarios. Our findings indicate that the behaviour of these algorithms varies across the three scenarios, emphasizing the need for detailed independent studies of each.

## 1 INTRODUCTION

The availability of ever-expanding datasets (Ardila et al., 2020; Wang et al., 2021b; Chan et al., 2021; Yang et al., 2024b) has facilitated the scaling of speech models (Radford et al., 2023; Zhang et al., 2024), leading to significant advancements in speech technology. Indeed, there is a growing trend towards training massive multilingual speech models on large amounts of data aggregated across multiple languages (Lugosch et al., 2021; Zhang et al., 2023). Given the substantial computational demands of these models, continual training has become crucial as new datasets for additional languages, domains, or demographics are introduced over time (Ardila et al., 2020; Gangwar et al., 2023). To address this, several continual learning techniques have emerged (Wang et al., 2024; Mundt et al., 2023), enabling efficient model updates with new data while preserving performance on previously learned tasks. These methods focus on three broad scenarios, *viz.*, *instance incremental learning*, *task incremental learning* and *domain incremental learning*.

Given the practical importance of Continual Learning (CL), several datasets and benchmarks have been proposed to evaluate the effectiveness of CL methods. However, most of these datasets, such as permuted MNIST (Goodfellow et al., 2014), Split-MNIST (Zenke et al., 2017), and Split-CIFAR (Krizhevsky et al., 2009), are synthetically derived from pre-existing datasets that were not incrementally collected. Since the original datasets were available all at once, there are no natural episodes, and for CL evaluation, episodes are artificially created by arbitrarily dividing the data. This differs significantly from how data arrives episodically in real-world scenarios, rendering these datasets inadequate for evaluating CL methods in such settings. More recently, benchmarks grounded in real-world scenarios, such as CLEAR (Lin et al., 2021), Visual Domain Decathlon

---

[1]Nirantar in Hindi means continual

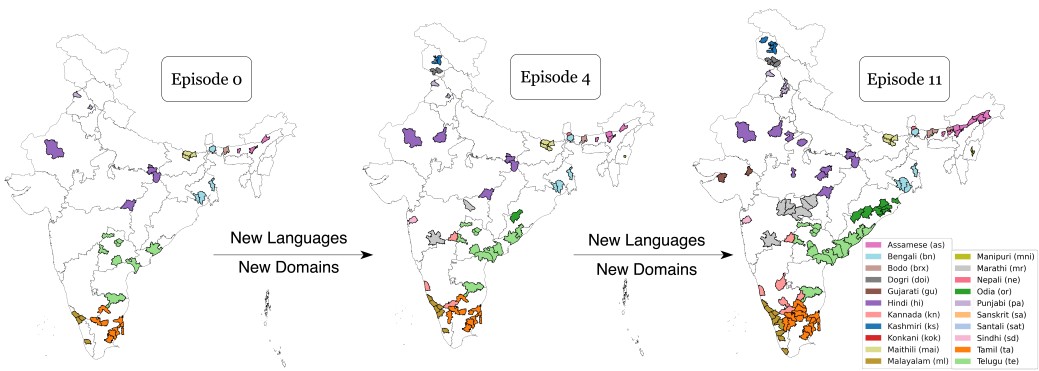

Figure 1: Illustration of Language-Incremental Domain-Incremental Learning: A practical scenario showing the addition of both new languages and domains in each episode of speech data collection. Our proposed episode timeline consists of a sequence of 208 domains across 22 languages.

(Rebuffi et al., 2017), Natural Language Decathlon (McCann et al., 2018), and CLIF (Jin et al., 2021), have been introduced to assess CL techniques. However, these benchmarks typically focus exclusively on either task-incremental learning or domain-incremental learning, and do not simultaneously address both or their combination.

In this work, we consider a practical on-ground speech data collection project for low-resource Indian languages, called IndicVoices (Javed et al., 2024b). This project aims to collect a representative and inclusive multilingual speech dataset covering 22 Indian languages and participants from 400 districts across the country. Data collection happens in batches and is coordinated by a team spread across the country. Specifically, at any given time, one or more districts corresponding to one or more of the 22 languages are identified. Following this, participants from the given district are solicited and asked questions specific to the district, local customs, and their interests. A total of around 20 to 50 hours of data is collected from each district, covering read, extempore, and conversational data on a random subset of topics, domains, and conversational scenarios relevant to that language and district. Each district serves as a domain due to its unique colloquial vocabulary, accents, and interests of local speakers. For example, a participant in Srinagar in northern India may talk about snow-capped mountains, whereas a participant in Assam in northeastern India may talk about tea plantations. Even for a given language, the choice of vocabulary, accents, topics of interest (farming, education, politics, entertainment, travelling, etc.) varies from one district to another.

The episodic nature of the data, with periodic gaps between batches that change language and domain distribution, provides an ideal setting for training and evaluating continual learning methods. Exploiting this natural episodic inflow of data, we create Nirantar, a realistic data framework for training and evaluating CL methods in three different scenarios: Language-Incremental (LIL), Domain-Incremental (DIL), and Language-Incremental Domain-Incremental Learning (LIDIL). The third scenario is novel as shown in Figure 1, and has not been studied in previous works. Nirantar contains a total of 3250 hours of human-transcribed speech data, of which 1530 hours was derived from the training set of the IndicVoices dataset (Javed et al., 2024b) and the remaining 1720 hours were newly collected as a part of this work following the exact same procedure as IndicVoices. The training data is divided into 12 episodes, each containing new languages, new domains, or both. The evaluation data contains 15 minutes of diverse data for each domain and language pair. We intend to maintain this as a live, evolving benchmark by continuously adding 15 minute samples to our test set as more data is collected. Furthermore, given that the test data is sampled at the district level, it naturally allows evaluation in an episodic setting.

We evaluate several existing continual learning (CL) approaches on the Nirantar benchmark, including replay-based methods, such as Experience Replay (Rolnick et al., 2019) and regularization-based methods, such as Elastic Weight Consolidation (Zhou & Cao, 2021) and Memory-aware Synapse (Aljundi et al., 2018). We observe that these approaches demonstrated varying performance across the three continual learning scenarios. This variability suggests that current techniques may not be universally effective, highlighting the need for more robust approaches that can consistently per-

form well across diverse multilingual and multidomain settings. We also make a key observation regarding architecture-based methods for CL. We found that these methods, which require adding parameters to the backbone, are impractical in real-world scenarios involving multiple languages and domains. Specifically, the addition of each new language (22 in our case) and each new domain (208 in our case) necessitates introducing a new adapter to the model. Over time, this leads to excessive complexity and model bloat, rendering such popular methods infeasible in real-world settings like Nirantar.

To encourage further research, all code, data, and models resulting from this work will be publicly available under the CC-BY-4.0 license. We would like to highlight that the 22 languages covered in Nirantar belong to 4 different language families, with good linguistic diversity. We focus our case study on Indian languages as they provide a good mix of medium-resource (eg, Tamil, Bengali), low-resource (eg. Marathi, Urdu, Konkani) and extremely low-resource (eg. Sindhi, Manipuri) languages. Given this, we believe that the observations made using Nirantar will be relevant for other low-resource language groups, and a broad set of language families as well.

## 2 RELATED WORK

Prior work in CL is broadly categorized into three types: regularization-based methods, replay-based methods, and architecture-based methods (Wang et al., 2023). Regularization-based methods, such as Elastic Weight Consolidation (EWC) (Zhou & Cao, 2021) and Memory-aware Synapses (MAS) (Aljundi et al., 2018), constrain large updates to model weights. Replay-based methods like Experience Replay (ER) and its variants (Rolnick et al., 2019) store past examples to mitigate forgetting, with enhancements such as Dark Experience Replay (DER) (Buzzega et al., 2020) applying knowledge distillation to stored examples. Averaged Gradient Episodic Memory (A-GEM) (Chaudhry et al., 2019) modifies gradients to minimize interference between new and old tasks. Architecture-based methods like Progressive Neural Networks (PNNs) (Rusu et al., 2016) and PackNet (Mallya & Lazebnik, 2018) allocate parameters for new tasks while preserving old ones.

**Continual learning in ASR.** In ASR, Continual Learning (CL) has primarily been studied in two settings: Language-Incremental Learning and Domain-Incremental Learning (van de Ven et al., 2022). For instance, Sadhu & Hermansky (2020) propose decomposing a DNN ASR system into sub-models specific to each domain, while Chang et al. (2021) trains a monolingual hybrid CTC-transformer model to adapt to new data distributions. These studies mainly focus on monolingual ASR with a domain incremental setup. In contrast, CL-MASR (Libera et al., 2023) explores various CL strategies in a multilingual setup, examining the potential of large-scale pretrained models in a language (task) incremental setting. Despite these advancements, there has been limited attention to continually updating models in settings that mimic real-world data collection scenarios. Our work offers a more broader playground for assessment of multilingual models by studying all three scenarios of Language-Incremental Learning (LIL), Domain-Incremental Learning (DIL), and Language-Incremental Domain-Incremental Learning (LIDIL).

**Continual learning benchmarks.** To the best of our knowledge, we are the first to introduce Continual Learning with new languages and new domains for ASR. A similar scenario termed new instances and new classes (NIC) (Lomonaco & Maltoni, 2017; Ceccon et al., 2024) exists but our work adapts it uniquely to the ASR domain by providing a framework that handles continual learning challenges specific to multilingual and multi-domain ASR systems. This benchmark facilitates the comprehensive evaluation of ASR models under more realistic and dynamic conditions, thereby pushing the boundaries of current continual learning research in ASR.

## 3 NIRANTAR: CONTINUAL LEARNING ON REAL-WORLD SPEECH DATA

In this section, we introduce Nirantar, a playground for continual learning in Automatic Speech Recognition (ASR) with new languages and domains. We also introduce definitions that will be referenced throughout the remainder of this paper.

## 3.1 DEFINITIONS

**Data Batch** ($B$): A data batch represents a unit of data collection resulting from a single data gathering activity for a specific domain $d$ of a language $l$, drawn from a set of domains $\mathcal{D}$ across a collection of languages $\mathcal{L}$. It is represented as an ordered tuple $B = (l, d)$, where $l \in \mathcal{L}$ and $d \in \mathcal{D}$. In ASR, a data batch consists of a set of $(x, y)$ pairs, where $x$ denotes the raw speech signal and $y$ represents the corresponding transcript.

**Episode** ($E$): An episode may involve a single data batch ($B$) or multiple data batches. Typically, the collection of several data batches occurs in parallel. This is represented by a data collection episode $E$, which is defined as a set of data batches, as follows:

$$E = \{(l, d) \mid l \in \mathcal{L}, d \in \mathcal{D}\} \tag{1}$$

**Timeline** ($T$): A timeline $T$ is defined as a sequence of episodes, represented as follows:

$$T = \langle E_0, E_1, \ldots, E_t, \ldots, E_\tau \rangle \tag{2}$$

where $t$ denotes a time step within the timeline and $\tau$ represents the total number of episodes.

**Model** ($m$): A model $m$ is a learnt mapping $y = m(x)$ by training on a collection of data batches.

**Continual Learning Method** ($c$): Given a timeline $T$, and a base model $m_0$ obtained by training on $E_0$, the continual learning method $c(\cdot)$ produces the model $m_\tau$ iteratively, as follows -

$$m_t = c(E_t, m_{t-1}), \quad 1 \leq t \leq \tau \tag{3}$$

### 3.1.1 CONTINUAL LEARNING SCENARIOS

We now briefly discuss the three continual learning scenarios

**Language Incremental Learning (LIL)**: In the Language-Incremental Learning (LIL) scenario, a new language is added in each episode. Specifically, for a given time step $t$, an episode $E_t$ consists of all data batches corresponding to a language $L_t$, as shown below-

$$E_t = \{(L_t, d) \mid d \in \mathcal{D}\}, \quad \forall\, t \in \tau, L_t \in \mathcal{L} \tag{4}$$

**Domain Incremental Learning (DIL)**: In the Domain-Incremental Learning (DIL) scenario, new domains are added in each episode. Specifically, all languages are seen at $E_0$, as shown below -

$$E_0 = \{(l, d) \mid \cup\, l = \mathcal{L}\} \tag{5}$$

This ensures that no new languages are added in $E_t$ when $1 \leq t \leq \tau$, only new domains are added.

**Language-Incremental Domain-Incremental Learning (LIDIL)**: In the LIDIL scenario, our evaluation framework comprises of a episode that contains both new languages and new districts, as shown in Equation 1. Here, any random collection of data batches forms an episode, and any random sequence of episodes forms a timeline.

## 3.2 DATASET DESCRIPTION

We build on top of recently released IndicVoices dataset (Javed et al., 2024b), which represents one of the largest efforts to collect speech datasets, covering India's 22 constitutionally recognized languages. It contains read, extempore and conversational data from a diverse set of speakers with fair representation across age groups, genders, educational backgrounds, locations and occupations. We further improve on IndicVoices to build Nirantar, to enable training and evaluation of ASR systems in a continual learning scenario. Specifically, apart from the initial 1530 hours released as part of IndicVoices, we collect an additional 1720 hours as a part of this work, using the exact same procedure as the original work. We collected the data in phases with each phase involving collection of data batches in parallel from one or more districts for one or more languages. Our team of coordinators visited each district, and mobilised around 100-150 participants with the help of local partners. After taking consent from the participants and appropriately compensating them for

Table 1: Number of hours (#H), speakers (#Sp), and domains (#D) in Nirantar, along with the ISO codes for the languages.

| | iso | #H | #Sp | #D | | iso | #H | #Sp | #D |
|---|---|---|---|---|---|---|---|---|---|
| **Assamese** | as | 241 | 985 | 14 | **Manipuri** | mni | 42 | 166 | 3 |
| **Bengali** | bn | 209 | 733 | 11 | **Marathi** | mr | 118 | 447 | 10 |
| **Bodo** | brx | 291 | 1061 | 4 | **Nepali** | ne | 252 | 780 | 4 |
| **Dogri** | doi | 116 | 495 | 5 | **Odia** | or | 124 | 473 | 9 |
| **Gujarati** | gu | 20 | 72 | 4 | **Punjabi** | pa | 124 | 344 | 6 |
| **Hindi** | hi | 138 | 490 | 12 | **Sanskrit** | sa | 70 | 222 | 17 |
| **Kannada** | kn | 96 | 530 | 13 | **Santali** | sat | 164 | 433 | 8 |
| **Konkani** | kok | 103 | 245 | 4 | **Sindhi** | sd | 27 | 240 | 4 |
| **Kashmiri** | ks | 106 | 515 | 10 | **Tamil** | ta | 238 | 1242 | 19 |
| **Maithili** | mai | 248 | 726 | 9 | **Telugu** | te | 221 | 767 | 28 |
| **Malayalam** | ml | 170 | 504 | 10 | **Urdu** | ur | 124 | 564 | 10 |

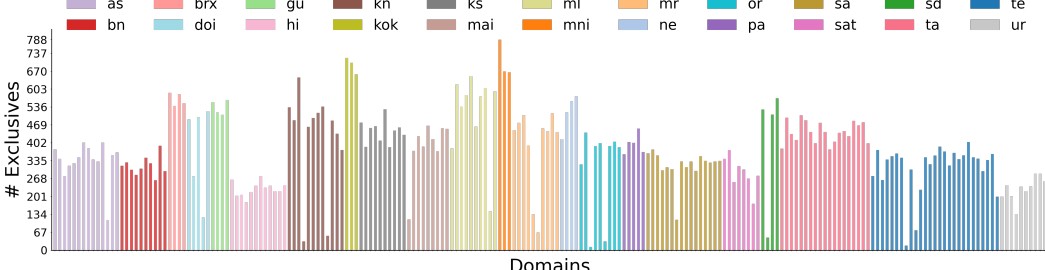

Figure 2: Number of unique words across each of the domains (districts) for all 22 Indian languages

their time, the coordinators recorded their (i) responses to tailored questions based on their topics of interest (ii) simulated interactions with voice assistants for everyday tasks like hailing a cab, making online payments, ordering food, etc. and (iii) two-party telephony interactions with other paid participants. The data was then transcribed with the help of an in-house team of transcribers comprising of makers, checkers and super-checkers to ensure quality.

Data collected from each district is treated as a batch and several data batches are aggregated to form a data episode. Each episode thus contains data from one or more languages consisting of one or more districts. Here, we consider each district as a new domain as the data characteristics vary from one district to another due to variation in accents, colloquial vocabulary, topics on interest and responses to questions which are specific to the given district. For example, as shown in Figure 2, the vocabulary usage changes across districts as indicated by the number of unique words added in each new district (each color corresponds to a different language). Nirantar thus leverages the natural influx of audio data in batches and splices the audio speech data across multiple timelines, one each for LIL, DIL, LIDIL. The creation of the timelines is highlighted in Section 3.3. Nirantar contains 3250 hours of data covering 208 districts across 22 Indian languages. Table 1 presents the statistics of data across languages. For creating the test data, we sample a maximum of 15 minutes from each of the domains resulting in a total of 50 hours across languages. Since the test data contains samples from every district, we can evaluate the forward and backward transfer of CL approaches.

### 3.3 CONTINUAL LEARNING PLAYGROUND

The Nirantar playground comprises three distinct timelines corresponding to LIL, DIL and LIDIL scenarios respectively. Table 2 outlines the distribution of data batches. Next, we present the process of creation of the timelines.

**Base episode** ($E_0$): In a practical scenario, the base model ($m_0$) will be trained after a seed amount of data is collected. We consider a good starting point for the base episode ($E_0$) when data batches are collected for half of the languages and half of the domains in each language. With this in mind, for LIDIL, we select the 11 languages having the largest number of hours in Table 1, and randomly

Table 2: Statistics showing the number of districts per language and the corresponding total number of hours (# H) of data for each episode (Ep) across LIL, DIL, and LIDIL settings. Each row represents an episode.

| Ep | as | bn | brx | doi | gu | hi | kn | kok | ks | mai | ml | mni | mr | ne | or | pa | sa | sat | sd | ta | te | ur | #H |
|---|---|---|---|---|---|---|---|---|---|---|---|---|---|---|---|---|---|---|---|---|---|---|---|
| **LIL** | | | | | | | | | | | | | | | | | | | | | | | |
| 0 | 14 | 11 | 4 | - | - | 12 | - | - | - | 9 | 10 | - | - | 4 | - | 6 | - | 8 | - | 19 | 28 | - | 2248 |
| 1 | - | - | - | 5 | - | - | - | - | - | - | - | - | - | - | - | - | - | - | - | - | - | - | 113 |
| 2 | - | - | - | - | - | - | 10 | - | - | - | - | - | - | - | - | - | - | - | - | - | - | - | 121 |
| 3 | - | - | - | - | 4 | - | - | - | - | - | - | - | - | - | - | - | - | - | - | - | - | - | 100 |
| 4 | - | - | - | - | - | - | - | - | - | - | - | - | - | - | - | - | - | - | - | - | - | 10 | 121 |
| 5 | - | - | - | - | - | - | - | 4 | - | - | - | - | - | - | - | - | - | - | - | - | - | - | 115 |
| 6 | - | - | - | - | - | - | - | - | - | - | - | - | - | - | 9 | - | - | - | - | - | - | - | 94 |
| 7 | - | - | - | - | - | - | - | - | - | - | - | - | 10 | - | - | - | - | - | - | - | - | - | 40 |
| 8 | - | - | - | - | - | - | 13 | - | - | - | - | - | - | - | - | - | - | - | - | - | - | - | 68 |
| 9 | - | - | - | - | - | - | - | - | - | - | - | 3 | - | - | - | - | - | - | - | - | - | - | 26 |
| 10 | - | - | - | - | - | - | - | - | - | - | - | - | - | - | - | - | - | 17 | - | - | - | - | 103 |
| 11 | - | - | - | - | - | - | - | - | - | - | - | - | - | - | - | - | - | - | 4 | - | - | - | 19 |
| **DIL** | | | | | | | | | | | | | | | | | | | | | | | |
| 0 | 7 | 5 | 2 | 2 | 2 | 6 | 6 | 2 | 5 | 4 | 5 | 1 | 5 | 2 | 4 | 3 | 8 | 4 | 2 | 9 | 14 | 5 | 1610 |
| 1 | - | - | - | 1 | - | - | 3 | - | 1 | - | 1 | - | 1 | 1 | - | - | - | - | - | 1 | 1 | - | 244 |
| 2 | 1 | - | - | - | - | - | - | - | 1 | 1 | 1 | - | 2 | 1 | 1 | - | - | 2 | - | 1 | - | - | 153 |
| 3 | 1 | - | - | 1 | - | 1 | - | - | - | - | 1 | - | - | - | 1 | - | - | - | - | 1 | 3 | - | 104 |
| 4 | 1 | - | - | - | 1 | 1 | - | - | - | 1 | - | 1 | - | - | - | - | - | - | - | 1 | 3 | - | 36 |
| 5 | - | - | 1 | - | 1 | - | 1 | - | - | - | - | - | - | - | - | - | - | 1 | - | 1 | - | 1 | 125 |
| 6 | 1 | - | - | - | - | - | - | - | 1 | - | - | - | - | - | - | - | - | - | - | - | 1 | 1 | 120 |
| 7 | - | 1 | - | 1 | - | - | - | - | 1 | 1 | - | - | - | - | - | 2 | - | - | - | - | - | - | 114 |
| 8 | 1 | - | - | - | - | 2 | 1 | - | - | - | - | - | - | - | - | 1 | - | - | - | 1 | 1 | 1 | 51 |
| 9 | - | 1 | 1 | - | - | - | - | - | 1 | - | 1 | - | - | - | - | - | 1 | - | - | - | 1 | - | 436 |
| 10 | - | - | - | - | - | 1 | 1 | - | - | 1 | - | 1 | - | - | - | - | 1 | - | - | 2 | - | - | 135 |
| 11 | 2 | 4 | - | - | - | 2 | 1 | - | - | 3 | - | 1 | 1 | - | 3 | - | 7 | 4 | - | 3 | 3 | 2 | 42 |
| **LIDIL** | | | | | | | | | | | | | | | | | | | | | | | |
| 0 | 7 | 5 | 2 | - | - | 6 | - | - | - | 4 | 5 | - | - | 2 | - | 3 | - | 4 | - | 9 | 14 | - | 1041 |
| 1 | - | - | - | - | - | 1 | - | - | 3 | - | 1 | - | 1 | - | 2 | - | 2 | 1 | 1 | - | - | - | 120 |
| 2 | - | - | - | 2 | - | - | 2 | - | 1 | 1 | - | - | - | - | - | 2 | - | - | 1 | - | - | - | 149 |
| 3 | 1 | - | - | - | - | 1 | - | - | - | - | - | - | 1 | 1 | - | 3 | - | 1 | - | 2 | 1 | - | 89 |
| 4 | - | 1 | - | - | - | - | 2 | - | 1 | 2 | - | 1 | 1 | - | - | 1 | - | - | 1 | 3 | 1 | - | 210 |
| 5 | - | 1 | - | - | 1 | 1 | - | - | 1 | - | - | 1 | - | - | - | 3 | - | 1 | - | 2 | - | - | 177 |
| 6 | 2 | 1 | 1 | - | 1 | 1 | 1 | - | - | 2 | - | - | - | - | 1 | 1 | 1 | - | 2 | 1 | - | - | 117 |
| 7 | - | 2 | - | 2 | 1 | 1 | - | - | 1 | - | - | 1 | - | - | - | - | 1 | 1 | 1 | 2 | 2 | - | 348 |
| 8 | - | - | - | - | - | 2 | - | - | 1 | - | - | - | - | 3 | - | 1 | - | - | 1 | 1 | 1 | - | 245 |
| 9 | 1 | - | - | - | 1 | - | 1 | 1 | - | - | - | 4 | 1 | 2 | 1 | 2 | - | - | - | 1 | - | - | 339 |
| 10 | 3 | - | - | - | - | 1 | 2 | 2 | 1 | - | 2 | 1 | - | 1 | - | 1 | 1 | 1 | - | - | 2 | 3 | 140 |
| 11 | - | 1 | 1 | 1 | 1 | 1 | 1 | 1 | 3 | - | - | - | 1 | - | 1 | 1 | 1 | - | - | 4 | - | 1 | 194 |

sample half the number of domains in each of these languages to create $E_0$. For LIL, we start with the same set of 11 languages, having all domains of the respective languages. For DIL, we start with all 22 languages, and randomly sample half the number of domains in each language.

**Incremental episodes** ($E_{\tau \geq 1}$): We create timelines of length $\tau = 11$. For LIL, all data batches corresponding to one language are added in each episode. The order of the languages is chosen randomly. For DIL and LIDIL, each data batch is randomly assigned to an episode. This ensures uniform distribution of data batches across episodes, while still ensuring non-uniformity in number of training hours across episodes, as shown in Table 2.

The purpose of this playground is to find an optimal continual learning approach $c^*$ given a timeline $T$ and a model $m_0$. Specifically, $c^* = \min_{c \in \mathcal{C}} V(c \mid T, m)$, where $V$ is a verifier or a metric that evalutes the continual learning approach, and $\mathcal{C}$ is a family of continual learning approaches. We explore a set of continual learning approaches and a set of metrics in Section 4 of the paper.

## 4 EXPERIMENTAL SETUP

We now describe the experimental setup used for training various models and evaluating their performance on Nirantar.

## 4.1 CONTINUAL LEARNING METHODS

Referring to the recently released survey paper (Mundt et al., 2023), we note that there are three main categories of popular continual learning (CL) methods, *viz.*, replay based methods, regularisation based methods and architecture-based methods. After careful consideration, we find that, architecture-based methods are not suited for real-world scenarios like Nirantar. This is because they require adding parameters for each new language (22, in our case) and each new domain (208, in our case) leading to excessive complexity and significant model expansion as the number of episodes grows. Given these limitations of architecture-based approaches, in this work, we focus on widely adopted and scalable CL techniques involving replay-based and regularization-based strategies. Below, we list down all the approaches considered in this work.

**Incremental Finetuning (Inc. FT):** Given a base model $m_0$, we sequentially finetune models $m_{1 \leq t \leq \tau}$ using the data batches in $E_t$, and initializing the weights of $m_t$ using the trained model $m_{t-1}$.

**Joint Finetuning (Joint FT):** Similar to Incremental Finetuning, we sequentially finetune $m_{1 \leq t \leq \tau}$ by initializing the weights of $m_t$ using the trained model $m_{t-1}$, but by taking all data batches from $\bigcup_{i=0}^{t} \{E_i\}$.

**Elastic Weight Consolidation (EWC) (Zhou & Cao, 2021):** EWC performs regularization by preserving important parameters from previous episodes while adapting to new ones. It estimates parameter importance using the Fisher information matrix (F) and adds a penalty term to the loss function during training on the current task. This penalty term, controlled by hyperparameters $\lambda$ and $\alpha$, balances between adapting to new tasks and retaining old knowledge. Following Libera et al. (2023) we set $\lambda$ to 5 and $\alpha$ to 0.5.

**Experience Replay (ER) (Rolnick et al., 2019):** Experience replay is a replay-based approach that stores data from previous episodes in a memory buffer and replays them during the training of models on current episodes. Following Libera et al. (2023), we sample 3% of data across each episode.

**Memory-aware Synapse (MAS) (Aljundi et al., 2018):** Like EWC, this method confines large model updates to weights. However, unlike the Fisher information matrix, it assesses parameter importance using the average magnitude of gradients of the squared L2 norm of the learned function. Following Libera et al. (2023), we set $\alpha$ and $\lambda$ to 1 and 0.5, respectively. These values determine the relative strength of the regularization term and the influence of previous tasks on updating parameter importance.

## 4.2 TRAINING

We train Conformer-L (Gulati et al., 2020) models, consisting of 120M parameters, as the encoder, with a hybrid CTC-RNNT (Noroozi et al., 2023) decoder. The model has 17 conformer blocks with 512 as the model dimension. The output vocabulary is of size 256 per language, and is created by a Byte-Pair-Encoding (BPE) tokenizer. Each language consists of a separate decoder head. All our models are trained using the NeMo (Kuchaiev et al., 2019) library. The base models $m_0$ and the Joint FT models were trained for 150,000 steps with a constant learning rate of 0.0001. Due to the skew in data distribution across languages in our joint multilingual setup, we found temperature sampling to be crucial for model convergence. We trained the incremental models for 30,000 steps with half the learning rate. We trained the models using the Adam optimizer with an effective batch size of 8 audios per GPU. All experiments utilized a total compute of 240 GPU-hours on 8 40GB-A100 GPUs.

## 4.3 METRICS

To study and compare performance across different continual strategies, we follow Libera et al. (2023) and use the following metrics:

**Average MER**: Match Error Rate (MER) (Morris et al., 2004) measures the probability of match being incorrect between the predicted transcript and the ground truth transcript. The overall perfor-

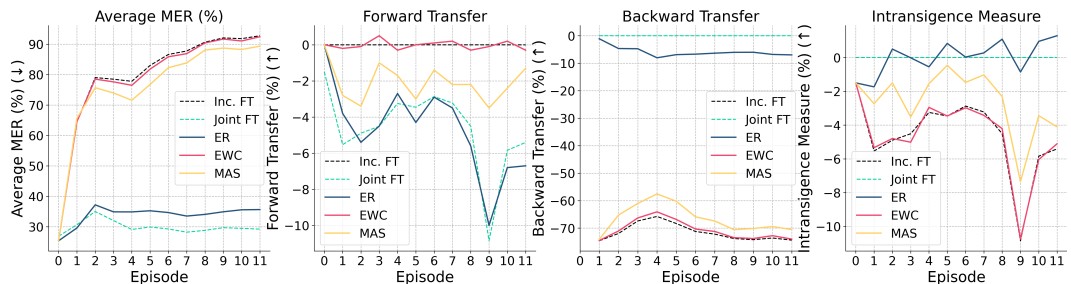

Figure 3: **Language-Incremental Learning (LIL)**: Comparison of various CL methods

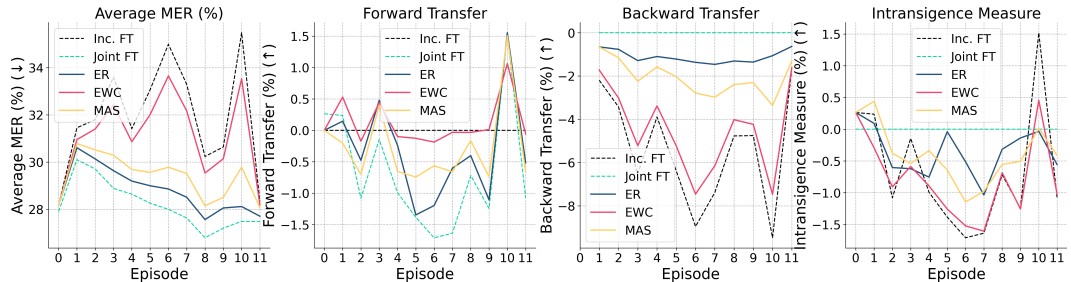

Figure 4: **Domain-Incremental Learning (DIL)**: Comparison of various CL methods

mance across all the seen episodes is calculated by

$$AMER_t = \frac{1}{t} \sum_{i=1}^{t} MER_{t,i}, \quad t \in [0, \tau]$$

**Forward Transfer**: This metric aims to capture the influence of previous episodes on the model's performance on the current episode. Specifically, it aims to quantify if the model is able to use the knowledge from the previous episode to help in improving the performance on the test set corresponding to the current episode. This metric is denoted by FWT and given by the following equation:

$$FWT_t = MER_t^{inc.ft} - MER_{t,t}$$

**Backward Transfer**: This quantifies the detriment in the model's performance on the knowledge learned from the previous episodes while learning new tasks and is given by the following equation:

$$BWT_t = \frac{1}{t-1} \sum_{i=1}^{t-1} MER_{i,i} - MER_{t,i}, \quad t \in [1, \tau]$$

**Intransigence Measure**: It quantifies the plasticity of the models, which refers to the model's capacity to acquire new knowledge effectively, as given by the following equation:

$$IM_t = MER_{t,t} - MER_t^{jointft}$$

## 5    RESULTS AND DISCUSSIONS

### 5.1    COMPARISON OF CONTINUAL LEARNING METHODS ACROSS THE 3 SCENARIOS

Figures 3, 4 and 5 present the main results of our study, comparing three continual learning (CL) approaches — ER, EWC, and MAS — across three scenarios: LIL, DIL, and LIDIL.

**LIL:** Referring to Figure 3, we observe a steady increase in AMER as new languages are introduced for Incremental FT. This is undesirable and highlights the need for effective continual learning (CL) methods. Both regularization-based approaches, EWC and MAS, struggle to retain knowledge of

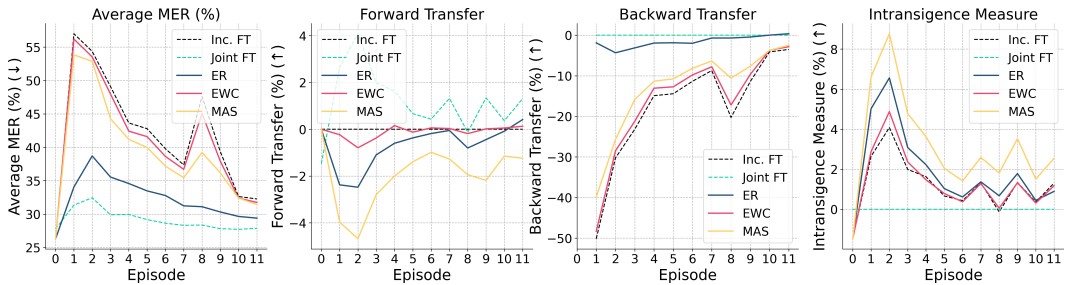

Figure 5: **Language-Incremental Domain-Incremental Learning (LIDIL)**: Comparison of various CL methods

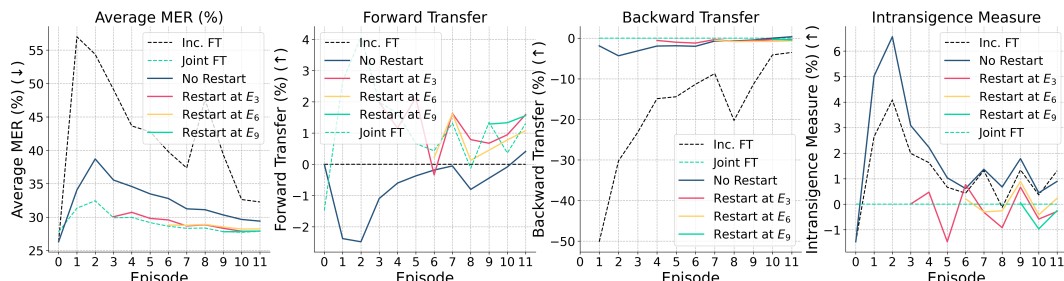

Figure 6: **ER with restarts for LIDIL**: Comparison across restarts from episodes 3, 6 and 9.

previously learned languages, as shown by the trends in the Forward Transfer across episodes. In contrast, ER significantly outperforms them, even with a buffer size of just 3%, demonstrating the importance of replay in LIL. While ER demonstrates strong backward transfer and positive intransigence, its poor forward transfer further emphasizes the need for CL approaches that better leverage knowledge from previous episodes. We also observe a sharp drop in the forward transfer and intransigence measures at episode 9. We hypothesize that this decline is due to the introduction of Manipuri, a Tibeto-Burman language with only 26 hours of data. The limited data and its notable differences from the Indo-Aryan and Dravidian language families observed in earlier episodes are likely factors contributing to this decline.

**DIL:** Referring to Figure 4, unlike LIL, we observe that AMER reduces over episodes for two methods, MAS and ER. The reduction of AMER over episodes could be attributed to (i) current CL methods being able to adapt better to new domains than to new languages, and (ii) the slightly favorable scenario in DIL, where the base model has already seen all the languages. This indicates the need of better base models to be used for CL. All CL approaches demonstrate good forward transfer and intransigence measure in DIL. The observed performance change of only 1.5% is due to the randomness in the order of incoming data batches. This indicates that knowledge from previous domains is indeed helpful for new domains. Although MAS performs significantly poor in LIL, we observe that it shows good Forward Transfer and Backward Transfer in DIL, showing that regularization-based methods are well suited for domain-incremental learning.

**LIDIL:** In Figure 5, we observe across all methods that the AMER first increases in the first 2 episodes similar to LIL, and then steadily decreases from episode 3 onwards, similar to DIL. This is due to the fact that many new languages are seen in the first 2 episodes, and the number of new languages gradually reduces after that. This demonstrates the unique hybrid nature of this newly introduced continual learning scenario that encompasses characteristics from both the aforementioned scenarios, *viz.*, LIL and DIL. We also observe that the backward transfer for EWC and MAS improves over time, unlike the other two paradigms, showing that the methods gradually adapt to previous tasks after addition of new languages and domains. All methods show a positive Intransigence Measure in LIDIL.

## 5.2 EFFECT OF RESTARTING

As observed in values of average MER in LIDIL for various CL methods, once the model training diverges in a certain episode, it is difficult for the model to catch up. In such cases, it is better to perform a Joint FT. To study this, we allow the CL methods to perform a 'restart' at episodes 3, 6 and 9. Specifically, at these episodes, we start with a base model which has been jointly trained on all data up to this point followed by continual training with ER for the remaining episodes. Figure 6 highlights the results for different restart points for the LIDIL scenario. As seen in Figure 6, restarting leads to more stable training across episodes, allowing the model to recover from earlier divergence. This shows that using a simple and practical technique of restarting, we get a performance which is as good as Joint FT. Specifically, ER restarted at any of these three episodes yielded results that match with the performance of Joint FT.

**Performance and Efficiency** While the AMER for the Jointly Fine-Tuned models is the lowest, these models are the least efficient in terms of computational resources, as they require retraining on each episode. Conversely, the AMER of Incremental models is the highest in each episode due to catastrophic forgetting. Models with restarts fall in between, and offer a tradeoff between performance and efficiency. For example, models restarted at episode 3 are more performant but less efficient than those restarted at episode 6.

While we understand that restarting essentially undermines the core principle of continual learning, we intentionally include this in our work to show that continual learning methods are still not competitive to restarting (Joint FT) in the LIDIL setting. We conduct this experiment to address a practical situation where training from scratch for each episode is infeasible; however, there is some additional computational budget available for a single restart.

## 6 CONCLUSION

We presented Nirantar, a novel data framework designed to facilitate training and evaluation of continual learning (CL) methods in multilingual and multidomain settings. This dataset contains 3250 hours of human-transcribed speech data, including 1720 hours released for this study, organized into 12 episodes featuring diverse language and domain combinations. Evaluations using established CL methods such as Elastic Weight Consolidation, Memory-aware Synapse, and Experience Replay highlight the utility of the dataset across Language-Incremental (LIL), Domain-Incremental (DIL), and Language-Incremental Domain-Incremental Learning (LIDIL) scenarios. All associated resources are available under a CC-BY-4 license to support further research in this area.

## 7 ETHICS

The data collection process follows the same guidelines as IndicVoices (Javed et al., 2024b) and was thoroughly reviewed and approved by the Institute Ethics Committee. Participants were fully informed about the collection, their involvement, and the use of their data, and their consent was obtained beforehand. They received compensation aligned with local daily wages for their time and effort. No PII data will be shared externally, and measures were implemented to anonymize and protect sensitive information. Project staff were also compensated appropriately. Nirantar will be released under the CC-BY-4.0 license, permitting commercial use.

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

# A APPENDIX

Table 3 presents a comparative overview of relevant datasets that can be used in LIL, DIL and LIDIL scenarios.

Table 3: Table comparing different publicly available dataset and their usability in different CL scenarios.

| Dataset | #Langs | #Domains (present in Metadata) | # Hours | Audio Source | Transcription | Supported scenario | | |
|---|---|---|---|---|---|---|---|---|
| | | | | | | LIL | DIL | LIDIL |
| LibriSpeech (lib, 2015) | 1 | - | 1000 | Audiobooks | Force Aligned | ✗ | ✗ | ✗ |
| GigaSpeech (Harte et al., 2023) | 1 | 23 | 10000 | YouTube | Force Aligned | ✗ | ✓ | ✗ |
| VoxPopuli(Wang et al., 2021a) | 16 | - | 1800 | Parliament Recordings | Force Aligned | ✓ | ✗ | ✗ |
| TED-LIUM(Hernandez et al., 2018) | 1 | - | 452 | TED talks | Force Aligned | ✗ | ✗ | ✗ |
| Spoken Wikipedia (Baumann et al., 2019) | 3 | - | 1005 | Crowdsourcing | Force Aligned | ✓ | ✗ | ✗ |
| Multilingual TEDx (Salesky et al., 2021) | 8 | - | 765 | TED Talks | Force Aligned | ✓ | ✗ | ✗ |
| Multilingual LibriSpeech (Pratap et al., 2020) | 8 | - | 44500 | Audiobooks | Force Aligned | ✓ | ✗ | ✗ |
| GigaSpeech 2 (Yang et al., 2024a) | 3 | - | 22015 | YouTube | Pseudolabelled | ✓ | ✗ | ✗ |
| Switchboard Corpus [2] | 1 | - | 260 | Human | Manual | ✗ | ✗ | ✗ |
| Common Voice 19 (Ardila et al., 2020) | 131 | - | 21594 | Human | Manual | ✓ | ✗ | ✗ |
| FLEURS (Conneau et al., 2022) | 102 | - | 1400 | Human | Manual | ✓ | ✗ | ✗ |
| MSR Srivastava et al., 2018 | 3 | - | 150 | Human | Manual | ✓ | ✗ | ✗ |
| OpenSLR Kjartansson et al., 2018 | 6 | - | 1247 | Human | Manual | ✓ | ✗ | ✗ |
| Crowdsourced Multispeaker Speech Dataset (He et al., 2020) | 6 | - | 35 | Human | Manual | ✓ | ✗ | ✗ |
| MUCS (Diwan et al., 2021) | 3 | - | 350 | Human | Manual | ✓ | ✗ | ✗ |
| IndicSUPERB (Javed et al., 2023a) | 12 | - | 1684 | Human | Manual | ✓ | ✗ | ✗ |
| Shrutilipi (Bhogale et al., 2023a) | 12 | - | 6457 | Newsonair | Force Aligned | ✓ | ✗ | ✗ |
| Graamvaani Bhanushali et al. (2022) | 1 | - | 108 | Human | Manual | ✗ | ✗ | ✗ |
| IIIS-Mile A et al. (2022a;b) | 2 | - | 500 | Human | Manual | ✓ | ✗ | ✗ |
| Kashmiri Data Corpus [3] | 1 | - | 1 | Human | Manual | ✗ | ✗ | ✗ |
| Vāksañcayah (Adiga et al., 2021) | 1 | - | 78 | Human | Manual | ✗ | ✗ | ✗ |
| The IIIT-H Indic Speech Databases (Prahallad et al., 2012) | 7 | - | 11 | Human | Manual | ✓ | ✗ | ✗ |
| Microsoft-IITB Marathi Speech Corpus (Abraham et al., 2020) | 1 | - | 109 | Human | Manual | ✗ | ✗ | ✗ |
| SMC Malayalam Speech Corpus [4] | 1 | 4 | 2 | Human | Manual | ✗ | ✓ | ✗ |
| IITM ASR Challange [5] | 3 | - | 690 | YouTube | Force Aligned | ✓ | ✗ | ✗ |
| NPTEL (Bhogale et al., 2023b) | 8 | - | 6400 | YouTube | Force Aligned | ✓ | ✗ | ✗ |
| IndicTTS (ind, 2016) | 13 | - | 225 | Human | Manual | ✓ | ✗ | ✗ |
| Svarah (Javed et al., 2023b) | 1 | 37 | 10 | Human | Manual | ✗ | ✓ | ✗ |
| SPRING-INX (R et al., 2023) | 10 | - | 3302 | Human | Manual | ✓ | ✗ | ✗ |
| SPIRE-SIES (Singh et al., 2023) | 1 | 13 | 23 | Human | Pseudolabelled | ✗ | ✓ | ✗ |
| Lahaja (Javed et al., 2024a) | 1 | 83 | 12.5 | Human | Manual | ✗ | ✓ | ✗ |
| **Nirantar** | 22 | 208 | 3250 | Human | Manual | ✓ | ✓ | ✓ |

---

[2]https://catalog.ldc.upenn.edu/LDC97S62

[3]https://openslr.org/122/

[4]https://blog.smc.org.in/malayalam-speech-corpus/

[5]https://sites.google.com/view/indian-language-asrchallenge/home

Figures 7 to 9 present the results for the original episodic sequence (Random Order 1) and two additional randomized sequences (Random Order 2 and Random Order 3) in the LIDIL scenario. The following lines list the original task order and two more permutations of it for the LIDIL scenario.

- Random Order 1: 0→1→2→3→4→5→6→7→8→9→10→11
- Random Order 2: 0→11→1→2→10→8→5→9→3→4→6→7
- Random Order 3: 0→8→6→7→9→4→5→1→2→3→11→10

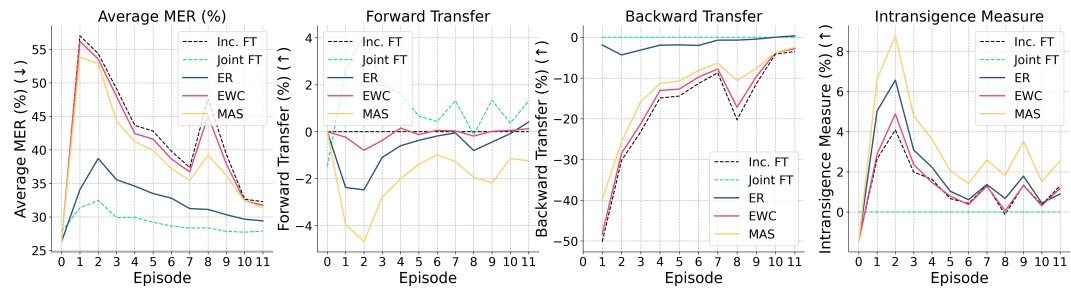

Figure 7: Random Order 1 for LIDIL Scenario.

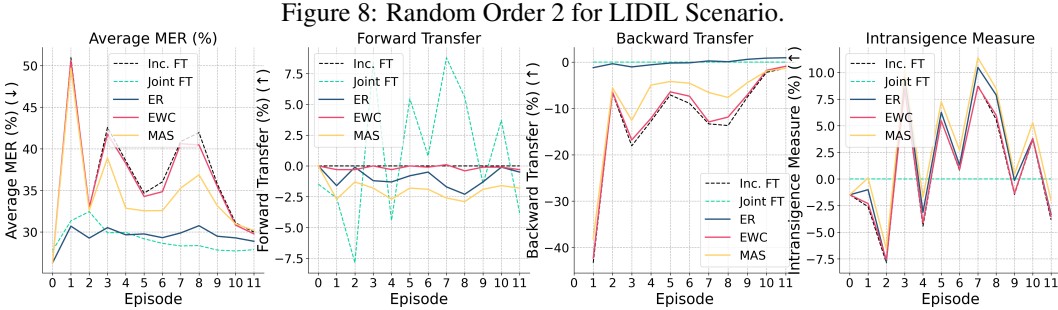

Figure 8: Random Order 2 for LIDIL Scenario.

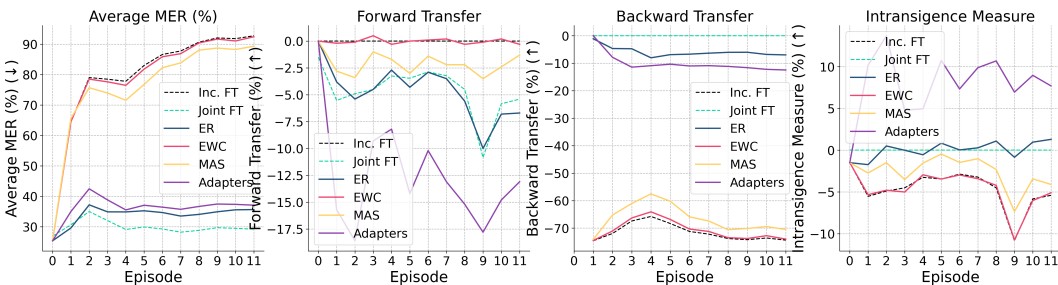

Figure 9: Random Order 3 for LIDIL Scenario

Figure 10 present the results LIL scenario involving adapters.

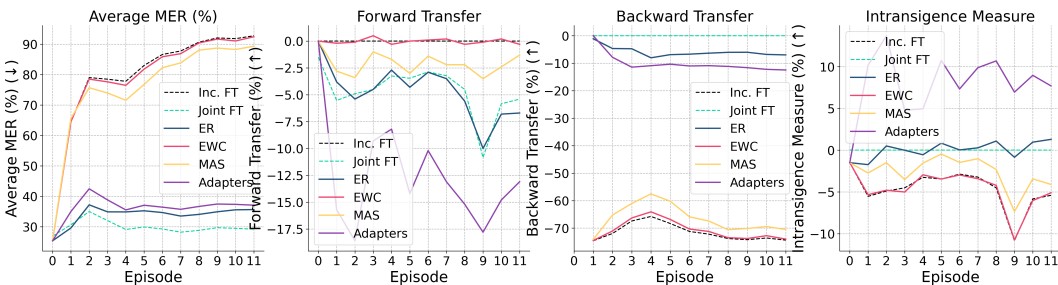

Figure 10: Results on LIL scenario using different CL methods, including adapters.

Figures 11 to 13 show the cross-lingual transfer of information for two language families, Indo-Aryan and Dravidian, in the LIDIL setting.

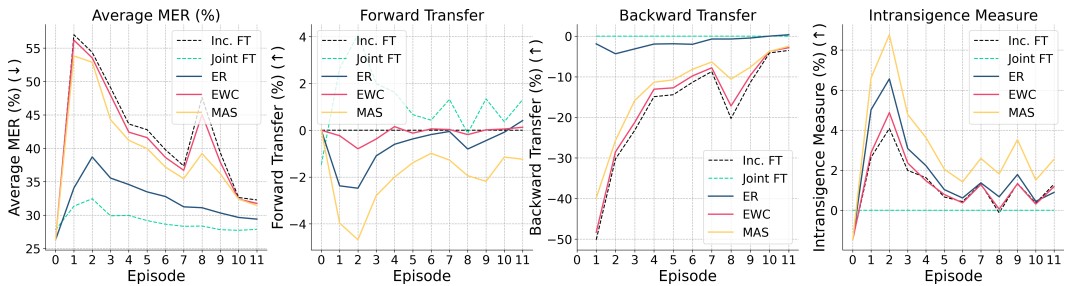

Figure 11: Comparison of different CL approaches for LIDIL scenario

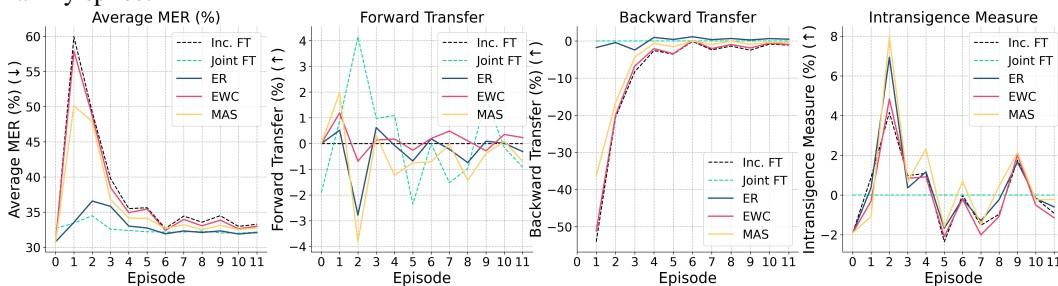

Figure 12: Comparison of different CL approaches for LIDIL scenario for IndoAryan language family splice.

Figure 13: Comparison of different CL approaches for LIDIL scenario for Dravidian language family splice.

Figures 14 to 15 illustrate how the domains and vocabulary evolve over episodes.

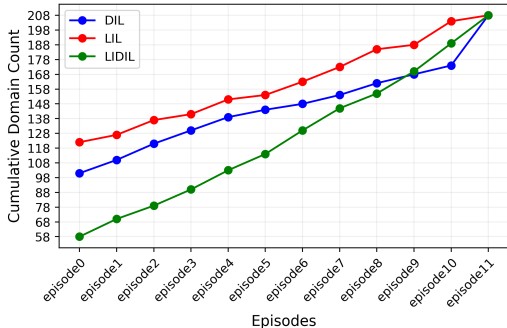

Figure 14: Figure showing the cumulative improvement of domains across episodes.

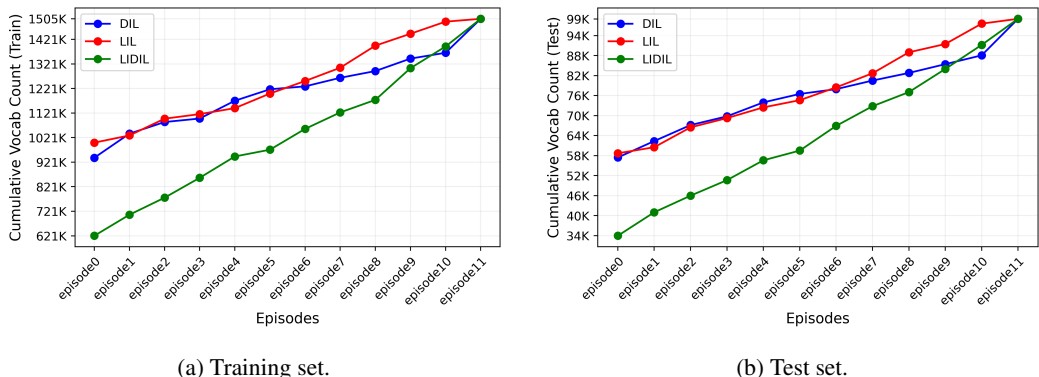

(a) Training set.                    (b) Test set.

Figure 15: Comparison of cumulative vocabulary improvement across episodes for training and test sets.

Table 4: Table comparing different publicly available dataset and their usability in different CL scenarios

| Language Code | Domain/ District | Train (hours) | Test (minutes) | WER (on Test) | Episodic presence | | |
|---|---|---|---|---|---|---|---|
| | | | | | LIL | DIL | LIDIL |
| as | Barpeta | 12 | 15.1 | 22.3% | episode0 | episode0 | episode0 |
| as | Biswanath | 18 | 15.1 | 16.3% | episode0 | episode0 | episode10 |
| as | Charaideo | 7.7 | 15.0 | 18.9% | episode0 | episode0 | episode9 |
| as | Darrang | 14.3 | 15.1 | 24.4% | episode0 | episode3 | episode6 |
| as | Dhemaji | 19.4 | 15.1 | 17.6% | episode0 | episode8 | episode10 |
| as | Dibrugarh | 17.6 | 15.0 | 19.1% | episode0 | episode2 | episode10 |
| as | Kamrup Metropolitan | 21.8 | 15.0 | 21.8% | episode0 | episode11 | episode0 |
| as | Lakhimpur | 38.6 | 15.0 | 22.1% | episode0 | episode11 | episode0 |
| as | Morigaon | 20.3 | 15.2 | 25.6% | episode0 | episode0 | episode0 |
| as | Nagaon | 18.7 | 15.1 | 23.3% | episode0 | episode0 | episode0 |
| as | Nalbari | 24.9 | 15.0 | 25.5% | episode0 | episode4 | episode0 |
| as | Sivasagar | 0.6 | 5.8 | 18.5% | episode0 | episode0 | episode0 |
| as | Sonitpur | 17.6 | 15.1 | 17.6% | episode0 | episode0 | episode3 |
| as | Tinsukia | 5.3 | 15.0 | 18.2% | episode0 | episode6 | episode6 |
| bn | Jalpaiguri | 0.8 | 15.1 | 18.8% | episode0 | episode11 | episode7 |
| bn | Jhargram | 28.9 | 15.1 | 15.2% | episode0 | episode0 | episode11 |
| bn | Nadia | 24.1 | 15.0 | 17.7% | episode0 | episode11 | episode0 |
| bn | North 24 Parganas | 2.9 | 15.1 | 13.5% | episode0 | episode9 | episode0 |
| bn | Paschim Bardhaman | 31.7 | 15.0 | 15.7% | episode0 | episode11 | episode6 |
| bn | Paschim Medinipur | 29.8 | 15.0 | 16.6% | episode0 | episode7 | episode0 |
| bn | Purba Bardhaman | 24.7 | 15.0 | 17.9% | episode0 | episode0 | episode0 |
| bn | Purba Medinipur | 23.2 | 15.0 | 17.7% | episode0 | episode0 | episode5 |
| bn | Purulia | 19.5 | 15.2 | 16.6% | episode0 | episode0 | episode7 |
| bn | South 24 Parganas | 18.7 | 15.0 | 17.8% | episode0 | episode0 | episode4 |
| brx | Baksa | 51.3 | 15.1 | 26.3% | episode0 | episode9 | episode0 |
| brx | Chirang | 106.2 | 15.1 | 28.2% | episode0 | episode0 | episode11 |
| brx | Kokrajhar | 81.3 | 15.1 | 29.0% | episode0 | episode5 | episode0 |
| brx | Udalguri | 46.2 | 15.0 | 30.8% | episode0 | episode0 | episode6 |
| hi | Darbhanga | 3.4 | 15.1 | 13.5% | episode0 | episode0 | episode0 |
| hi | Balaghat | 7.6 | 15.1 | 16.7% | episode0 | episode0 | episode0 |
| hi | Bhopal | 26.9 | 15.2 | 12.6% | episode0 | episode0 | episode7 |
| hi | Gwalior | 2.2 | 15.0 | 15.2% | episode0 | episode3 | episode5 |
| hi | Jabalpur | 2.2 | 15.2 | 16.9% | episode0 | episode0 | episode11 |
| hi | Katni | 1.6 | 15.0 | 15.2% | episode0 | episode0 | episode10 |
| hi | Jaipur | 27.3 | 15.1 | 16.3% | episode0 | episode0 | episode1 |
| hi | Jodhpur | 25.2 | 15.1 | 17.4% | episode0 | episode4 | episode0 |
| hi | Karauli | 10.2 | 15.1 | 16.4% | episode0 | episode8 | episode6 |
| hi | Bhadohi | 2.2 | 15.1 | 17.0% | episode0 | episode11 | episode0 |
| hi | Mirzapur | 4.9 | 15.0 | 18.0% | episode0 | episode11 | episode0 |
| hi | Sonbhadra | 20.7 | 15.1 | 16.8% | episode0 | episode8 | episode0 |
| mai | Darbhanga | 34.6 | 15.0 | 30.7% | episode0 | episode11 | episode2 |
| mai | Begusarai | 0.3 | 5.5 | 32.5% | episode0 | episode11 | episode4 |
| mai | Madhubani | 33.3 | 15.0 | 32.0% | episode0 | episode2 | episode0 |
| mai | Muzaffarpur | 26.8 | 15.1 | 32.7% | episode0 | episode0 | episode5 |
| mai | Purnia | 40.9 | 15.1 | 41.0% | episode0 | episode0 | episode0 |
| mai | Saharsa | 26.8 | 15.2 | 32.3% | episode0 | episode4 | episode4 |
| mai | Samastipur | 7.3 | 15.0 | 40.4% | episode0 | episode11 | episode0 |
| mai | Sitamarhi | 32.7 | 15.4 | 38.8% | episode0 | episode0 | episode8 |
| mai | Supaul | 39.9 | 15.0 | 35.7% | episode0 | episode0 | episode0 |
| ml | Ernakulam | 0.1 | 11.7 | 32.3% | episode0 | episode10 | episode10 |
| ml | Kannur | 14.4 | 15.2 | 41.6% | episode0 | episode0 | episode0 |
| ml | Kasaragod | 11 | 15.0 | 36.4% | episode0 | episode0 | episode6 |
| ml | Kottayam | 12.4 | 15.1 | 39.4% | episode0 | episode0 | episode0 |
| ml | Kozhikode | 40.1 | 15.1 | 37.4% | episode0 | episode1 | episode0 |
| ml | Malappuram | 1.2 | 15.1 | 34.2% | episode0 | episode2 | episode1 |
| ml | Palakkad | 45.5 | 15.1 | 39.3% | episode0 | episode0 | episode10 |

| Language Code | Domain/ District | Train (hours) | Test (minutes) | WER (on Test) | Episodic presence | | |
|---|---|---|---|---|---|---|---|
| | | | | | LIL | DIL | LIDIL |
| ml | Thiruvananthapuram | 6.9 | 15.0 | 37.8% | episode0 | episode0 | episode6 |
| ml | Thrissur | 1.8 | 3.9 | 39.6% | episode0 | episode3 | episode0 |
| ml | Wayanad | 32.5 | 15.2 | 44.8% | episode0 | episode9 | episode0 |
| ne | Jalpaiguri | 22.5 | 15.2 | 20.6% | episode0 | episode0 | episode0 |
| ne | Alipurduar | 1.3 | 15.1 | 25.8% | episode0 | episode2 | episode0 |
| ne | Darjeeling | 109.8 | 15.1 | 17.1% | episode0 | episode0 | episode9 |
| ne | Kalimpong | 113.3 | 15.0 | 17.0% | episode0 | episode1 | episode3 |
| pa | Fatehgarh Sahib | 27.8 | 15.0 | 15.2% | episode0 | episode8 | episode0 |
| pa | Mohali | 34.5 | 15.0 | 11.7% | episode0 | episode0 | episode0 |
| pa | Patiala | 1.5 | 15.1 | 17.0% | episode0 | episode0 | episode11 |
| pa | Rupnagar | 30.5 | 15.0 | 13.5% | episode0 | episode7 | episode6 |
| pa | Shaheed Bhagat Singh Nagar | 27.5 | 15.0 | 12.3% | episode0 | episode7 | episode9 |
| sat | Jhargram | 22 | 15.1 | 29.2% | episode0 | episode11 | episode0 |
| sat | Paschim Bardhaman | 26.4 | 15.1 | 31.4% | episode0 | episode11 | episode10 |
| sat | Purba Bardhaman | 21.7 | 15.1 | 35.3% | episode0 | episode0 | episode0 |
| sat | Purulia | 6.3 | 15.0 | 46.7% | episode0 | episode11 | episode1 |
| sat | Bankura | 33.8 | 15.1 | 34.3% | episode0 | episode0 | episode7 |
| sat | Birbhum | 45.7 | 15.0 | 40.3% | episode0 | episode0 | episode0 |
| sat | Malda | 1.4 | 15.0 | 40.4% | episode0 | episode0 | episode6 |
| sat | Uttar Dinajpur | 2.9 | 15.0 | 47.6% | episode0 | episode11 | episode0 |
| ta | Ariyalur | 4.4 | 15.1 | 29.0% | episode0 | episode3 | episode11 |
| ta | Coimbatore | 12.9 | 15.1 | 36.3% | episode0 | episode11 | episode8 |
| ta | Cuddalore | 11.7 | 15.0 | 31.4% | episode0 | episode1 | episode0 |
| ta | Dharmapuri | 12.1 | 15.0 | 34.7% | episode0 | episode0 | episode6 |
| ta | Erode | 15.3 | 15.1 | 33.9% | episode0 | episode0 | episode0 |
| ta | Kallakurichi | 16 | 15.0 | 32.0% | episode0 | episode0 | episode7 |
| ta | Krishnagiri | 13.8 | 15.0 | 32.6% | episode0 | episode0 | episode0 |
| ta | Mayiladuthurai | 32.2 | 15.1 | 34.9% | episode0 | episode11 | episode2 |
| ta | Nagapattinam | 20.4 | 15.1 | 37.1% | episode0 | episode0 | episode0 |
| ta | Namakkal | 21 | 15.1 | 37.1% | episode0 | episode0 | episode0 |
| ta | Perambalur | 2.6 | 15.1 | 37.1% | episode0 | episode10 | episode4 |
| ta | Pudukkottai | 6 | 15.1 | 26.4% | episode0 | episode4 | episode0 |
| ta | Salem | 10.8 | 15.0 | 33.7% | episode0 | episode0 | episode6 |
| ta | Sivaganga | 15.1 | 15.1 | 35.2% | episode0 | episode8 | episode0 |
| ta | Thanjavur | 1.3 | 15.0 | 36.5% | episode0 | episode11 | episode0 |
| ta | Tiruchirappalli | 3.1 | 15.1 | 38.4% | episode0 | episode5 | episode11 |
| ta | Tiruppur | 11.6 | 15.0 | 35.9% | episode0 | episode0 | episode11 |
| ta | Tiruvarur | 16.6 | 15.1 | 29.9% | episode0 | episode10 | episode11 |
| ta | Viluppuram | 5.8 | 15.1 | 27.7% | episode0 | episode0 | episode0 |
| te | Anakapalli | 1.1 | 15.3 | 20.0% | episode0 | episode11 | episode10 |
| te | Chittoor | 19.1 | 15.1 | 26.6% | episode0 | episode0 | episode0 |
| te | East Godavari | 14.9 | 15.1 | 28.1% | episode0 | episode11 | episode5 |
| te | Eluru | 10.6 | 15.1 | 21.4% | episode0 | episode0 | episode0 |
| te | Guntur | 8.3 | 15.1 | 20.7% | episode0 | episode1 | episode4 |
| te | Kakinada | 15.9 | 15.0 | 29.8% | episode0 | episode4 | episode0 |
| te | Konaseema | 12.8 | 15.0 | 16.9% | episode0 | episode6 | episode0 |
| te | Krishna | 2.1 | 3.2 | 23.1% | episode0 | episode0 | episode0 |
| te | N T Rama Rao | 4.3 | 15.3 | 27.8% | episode0 | episode3 | episode10 |
| te | Nellore | 5.6 | 15.1 | 34.3% | episode0 | episode2 | episode7 |
| te | Palnadu | 8.2 | 15.1 | 21.3% | episode0 | episode0 | episode0 |
| te | Sri Balaji | 18.7 | 15.1 | 32.0% | episode0 | episode4 | episode0 |
| te | Srikakulam | 10.8 | 15.0 | 29.6% | episode0 | episode0 | episode5 |
| te | Visakhapatnam | 2.3 | 15.1 | 29.8% | episode0 | episode0 | episode0 |
| te | Vizianagaram | 9.9 | 15.0 | 30.3% | episode0 | episode4 | episode3 |
| te | West Godavari | 4.7 | 15.2 | 24.7% | episode0 | episode9 | episode4 |
| te | Hyderabad | 16.7 | 15.0 | 31.3% | episode0 | episode0 | episode4 |
| te | Karimnagar | 0 | 1.6 | 7.8% | episode0 | episode8 | episode0 |
| te | Mahbubnagar | 1.1 | 15.2 | 21.4% | episode0 | episode3 | episode3 |
| te | Mancherial | 4.5 | 15.2 | 30.1% | episode0 | episode0 | episode8 |
| te | Medchal | 4.3 | 15.1 | 27.0% | episode0 | episode0 | episode7 |

| Language Code | Domain/ District | Train (hours) | Test (minutes) | WER (on Test) | Episodic presence | | |
|---|---|---|---|---|---|---|---|
| | | | | | LIL | DIL | LIDIL |
| te | Nalgonda | 7.5 | 15.1 | 29.8% | episode0 | episode0 | episode0 |
| te | Nirmal | 2.1 | 15.1 | 29.0% | episode0 | episode3 | episode0 |
| te | Ranga Reddy | 12.9 | 15.1 | 32.0% | episode0 | episode11 | episode9 |
| te | Sangareddy | 4.1 | 15.0 | 24.0% | episode0 | episode0 | episode0 |
| te | Vikarabad | 7.1 | 15.1 | 26.6% | episode0 | episode0 | episode0 |
| te | Yadadri Bhuvanagiri | 2.7 | 15.0 | 19.1% | episode0 | episode0 | episode6 |
| doi | Jammu | 12.2 | 15.1 | 30.4% | episode1 | episode3 | episode2 |
| doi | Kathua | 0.3 | 13.4 | 17.9% | episode1 | episode7 | episode7 |
| doi | Reasi | 55.1 | 15.2 | 30.1% | episode1 | episode0 | episode11 |
| doi | Samba | 0.8 | 4.9 | 22.4% | episode1 | episode0 | episode7 |
| doi | Udhampur | 45 | 15.0 | 35.6% | episode1 | episode1 | episode2 |
| sa | Chittoor | 3.9 | 15.1 | 19.6% | episode10 | episode11 | episode3 |
| sa | Bagalkot | 2 | 15.0 | 22.9% | episode10 | episode0 | episode10 |
| sa | Bangalore Rural | 0.6 | 15.1 | 21.4% | episode10 | episode10 | episode11 |
| sa | Bangalore Urban | 6.1 | 15.1 | 20.8% | episode10 | episode11 | episode5 |
| sa | Chikkamagaluru | 2.6 | 15.0 | 23.2% | episode10 | episode0 | episode2 |
| sa | Dakshina Kannada | 12.2 | 15.1 | 21.9% | episode10 | episode0 | episode3 |
| sa | Mysore | 3.8 | 15.0 | 17.3% | episode10 | episode11 | episode8 |
| sa | Shimoga | 4.3 | 15.1 | 20.3% | episode10 | episode0 | episode1 |
| sa | Udupi | 8.3 | 15.3 | 23.5% | episode10 | episode0 | episode9 |
| sa | Uttara Kannada | 11.4 | 15.1 | 22.3% | episode10 | episode0 | episode3 |
| sa | Nagpur | 0.6 | 15.1 | 17.4% | episode10 | episode11 | episode9 |
| sa | Jaipur | 2.6 | 15.2 | 24.7% | episode10 | episode11 | episode2 |
| sa | Coimbatore | 1.6 | 6.3 | 34.0% | episode10 | episode0 | episode1 |
| sa | Chennai | 3.3 | 15.1 | 24.0% | episode10 | episode9 | episode5 |
| sa | Hyderabad | 1.5 | 15.0 | 21.5% | episode10 | episode11 | episode6 |
| sa | Ranga Reddy | 0 | 15.0 | 21.7% | episode10 | episode0 | episode5 |
| sd | South Delhi | 0.1 | 2.0 | 21.6% | episode11 | episode0 | episode5 |
| sd | Surat | 2 | 15.0 | 20.0% | episode11 | episode2 | episode3 |
| sd | Mumbai Suburban | 3.5 | 15.0 | 20.8% | episode11 | episode0 | episode7 |
| sd | Thane | 20.5 | 15.1 | 23.2% | episode11 | episode2 | episode1 |
| ks | Anantnag | 11.2 | 15.1 | 43.5% | episode2 | episode0 | episode1 |
| ks | Bandipora | 3.7 | 15.2 | 30.8% | episode2 | episode0 | episode2 |
| ks | Baramulla | 11 | 15.1 | 45.8% | episode2 | episode0 | episode7 |
| ks | Budgam | 7.7 | 15.0 | 38.7% | episode2 | episode0 | episode11 |
| ks | Ganderbal | 16.5 | 15.1 | 34.8% | episode2 | episode0 | episode10 |
| ks | Kulgam | 16.2 | 15.0 | 45.6% | episode2 | episode7 | episode1 |
| ks | Kupwara | 11.8 | 15.1 | 42.7% | episode2 | episode6 | episode11 |
| ks | Pulwama | 2.5 | 15.2 | 36.4% | episode2 | episode1 | episode1 |
| ks | Shopian | 19.6 | 15.1 | 37.7% | episode2 | episode9 | episode11 |
| ks | Srinagar | 3.2 | 15.0 | 41.0% | episode2 | episode2 | episode4 |
| gu | Ahmedabad | 4.8 | 15.2 | 14.6% | episode3 | episode5 | episode9 |
| gu | Aravalli | 2.6 | 15.0 | 24.5% | episode3 | episode0 | episode11 |
| gu | Mehsana | 4.8 | 15.0 | 16.7% | episode3 | episode0 | episode6 |
| gu | Morbi | 6.9 | 15.2 | 20.4% | episode3 | episode4 | episode7 |
| ur | South Delhi | 12.4 | 15.0 | 13.1% | episode4 | episode11 | episode6 |
| ur | Central Delhi | 17 | 15.2 | 15.9% | episode4 | episode0 | episode10 |
| ur | Nashik | 14.1 | 15.0 | 13.6% | episode4 | episode0 | episode10 |
| ur | Hyderabad | 12.1 | 15.1 | 15.3% | episode4 | episode11 | episode7 |
| ur | Aligarh | 16.6 | 15.1 | 13.3% | episode4 | episode0 | episode7 |
| ur | Gautam Buddha Nagar | 18.5 | 15.2 | 13.4% | episode4 | episode8 | episode4 |
| ur | Ghaziabad | 2.6 | 14.3 | 19.2% | episode4 | episode0 | episode10 |
| ur | Lucknow | 18.2 | 15.1 | 14.1% | episode4 | episode0 | episode3 |
| ur | Mau | 3.6 | 15.1 | 13.8% | episode4 | episode5 | episode8 |
| ur | Shahjahanpur | 5.8 | 15.1 | 10.8% | episode4 | episode6 | episode11 |
| kok | Bardez | 33.1 | 15.1 | 32.3% | episode5 | episode0 | episode11 |
| kok | Canacona | 46.2 | 15.0 | 32.7% | episode5 | episode0 | episode9 |
| kok | Tiswadi | 20.3 | 15.1 | 27.9% | episode5 | episode7 | episode10 |
| or | Bhadrak | 2.2 | 15.0 | 18.6% | episode6 | episode0 | episode11 |
| or | Boudh | 12.5 | 15.0 | 28.8% | episode6 | episode11 | episode1 |

| Language Code | Domain/ District | Train (hours) | Test (minutes) | WER (on Test) | Episodic presence | | |
|---|---|---|---|---|---|---|---|
| | | | | | LIL | DIL | LIDIL |
| or | Cuttack | 0 | 0.7 | 37.5% | episode6 | episode3 | episode8 |
| or | Dhenkanal | 20.8 | 15.3 | 23.7% | episode6 | episode0 | episode10 |
| or | Jajpur | 15.3 | 15.1 | 22.0% | episode6 | episode0 | episode8 |
| or | Kalahandi | 0 | 1.2 | 42.9% | episode6 | episode11 | episode1 |
| or | Kandhamal | 21.6 | 15.1 | 21.3% | episode6 | episode0 | episode9 |
| or | Khordha | 26.4 | 15.1 | 21.0% | episode6 | episode11 | episode8 |
| or | Nayagarh | 22.3 | 15.0 | 22.9% | episode6 | episode2 | episode9 |
| mr | Nagpur | 15 | 15.0 | 18.9% | episode7 | episode0 | episode11 |
| mr | Thane | 4.3 | 15.1 | 16.5% | episode7 | episode2 | episode10 |
| mr | Akola | 24.9 | 15.2 | 17.1% | episode7 | episode10 | episode9 |
| mr | Amravati | 16.2 | 15.1 | 18.7% | episode7 | episode11 | episode1 |
| mr | Buldhana | 18.9 | 15.0 | 17.6% | episode7 | episode0 | episode9 |
| mr | Raigad | 0.6 | 6.8 | 18.0% | episode7 | episode0 | episode5 |
| mr | Solapur | 1.4 | 2.1 | 16.8% | episode7 | episode0 | episode4 |
| mr | Wardha | 2.5 | 15.1 | 16.4% | episode7 | episode1 | episode3 |
| mr | Washim | 7.5 | 15.1 | 22.3% | episode7 | episode0 | episode9 |
| mr | Yavatmal | 23.9 | 15.1 | 16.2% | episode7 | episode2 | episode9 |
| kn | Bangalore Rural | 5.7 | 15.0 | 34.2% | episode8 | episode10 | episode2 |
| kn | Bangalore Urban | 4 | 15.1 | 30.4% | episode8 | episode0 | episode8 |
| kn | Mysore | 1.2 | 2.1 | 43.6% | episode8 | episode0 | episode5 |
| kn | Shimoga | 17.6 | 15.0 | 22.2% | episode8 | episode11 | episode4 |
| kn | Udupi | 1.6 | 13.4 | 29.2% | episode8 | episode0 | episode2 |
| kn | Bidar | 8.1 | 15.0 | 43.6% | episode8 | episode1 | episode4 |
| kn | Chamarajanagar | 1.5 | 1.3 | 29.2% | episode8 | episode0 | episode3 |
| kn | Chikkaballapur | 8.3 | 15.1 | 22.2% | episode8 | episode8 | episode6 |
| kn | Chitradurga | 10.9 | 15.0 | 29.1% | episode8 | episode5 | episode11 |
| kn | Davanagere | 8.9 | 15.1 | 30.6% | episode8 | episode0 | episode10 |
| kn | Kolar | 14 | 15.2 | 23.8% | episode8 | episode0 | episode8 |
| kn | Tumkur | 11.4 | 15.0 | 26.9% | episode8 | episode1 | episode9 |
| mni | Imphal West | 18.6 | 15.1 | 21.6% | episode9 | episode4 | episode7 |
| mni | Kakching | 3.7 | 15.0 | 37.3% | episode9 | episode0 | episode10 |
| mni | Thoubal | 18.3 | 15.1 | 21.8% | episode9 | episode11 | episode4 |

