# OpenReview forum: "NIRANTAR: Continual Learning with New Languages and Domains on Real-world Speech Data"
_ICLR.cc/2025/Conference — Submitted to ICLR 2025_

### Official Review · Reviewer_YNAF · 2024-11-02

**Soundness:** 2
**Presentation:** 2
**Contribution:** 3
**Rating:** 5
**Confidence:** 4

**Summary:**

The paper introduce Nirantar, a large-scale dataset designed for continual learning (CL) in ASR, spanning 22 Indian languages and over 3,250 hours of human-transcribed speech. This dataset includes diverse domains, represented by 208 districts across India, to model real-world multilingual, multidomain CL challenges.

Nirantar's primary contributions include establishing 3 CL scenarios—Language-Incremental Learning (LIL), Domain-Incremental Learning (DIL), and Language-Incremental Domain-Incremental Learning (LIDIL). The LIDIL scenario is novel, introducing both new languages and domains across episodes, a setup that reflects real-world data inflow patterns and has not been previously explored in CL.

To validate the dataset's utility, the authors benchmark several existing CL techniques (including Elastic Weight Consolidation, Experience Replay, and Memory-Aware Synapse) across the 3 scenarios. Experiment results suggest potential areas for improvement in CL approaches for multilingual, multidomain ASR.

**Strengths:**

### Originality
The Nirantar dataset presents a valuable resource for the speech research communities. With 22 Indian languages and over 3,250 hours of human speech, including 1,720 hours newly released as part of this work, and human-annotated transcriptions, the dataset spans a wide variety of topics, domains, and conversational scenarios. The paper introduces a novel Language-Incremental Domain-Incremental Learning (LIDIL) scenario, expanding beyond the traditional Language-Incremental (LIL) and Domain-Incremental (DIL) scenarios that form the three core scenarios of the Nirantar dataset. The inclusion of LIDIL sets a new benchmark for future studies in multilingual and multidomain CL under real-world scenarios.

### Quality
The paper is structured-well and offer a clear overview of the data collection, scenarios, and experimental evaluations on existing CL approaches.

### Clarity
The authors present a clear breakdown of 3 CL scenarios.

### Significance
The dataset’s breadth across 22 Indian languages and 208 districts makes it a valuable resource for developing speech models adaptable to language and domain shifts.

**Weaknesses:**

The paper does not report pre-CL performance metrics, such as Word Error Rate (WER,) for individual episodes across languages and domains, which limits understanding of Nirantar’s distinctiveness as a CL dataset. Specifically, existing ASR datasets like GigaSpeech 2 could be reorganized to create a comparable multi-domain, multilingual LIDIL dataset, so to strengthen its case as a novel CL dataset, additional evidence is needed. For instance, including initial statistics per language, domain, and episode would provide insights into each batch's difficulty across diverse linguistic and regional characteristics, highlighting its uniqueness as a new ASR CL dataset. Additionally, including initial ASR WER would help clarify Nirantar's quality as an ASR dataset.

**Questions:**

### Dataset Quality:
1. Could you provide details on the quality assurance procedures and any pre-CL performance metrics, such as initial Word Error Rate (WER), to validate transcription accuracy across languages/domains/episodes? Understanding these methods and statistics would clarify the dataset’s reliability and distinctiveness for continual learning applications.

2. Could you elaborate on the procedures to ensure topic and domain diversity across districts, and how this diversity is measured or validated in different episodes? These insights would help demonstrate the dataset’s capacity to represent unique linguistic and regional characteristics, supporting its role as a comprehensive ASR CL dataset.

---

> ### Author Response · Authors · 2024-11-22
> **Response to reviewer YNAF**
>
> We sincerely thank the reviewer for their useful feedback, please find our responses below.
>
>     Items marked with ** indicate hyperlinks
>
> **W1 (partial): ‘existing ASR datasets like GigaSpeech 2 could be reorganized’**
>
> While we acknowledge the existence of certain datasets that can be repurposed for continual learning in LIL or DIL scenarios, we could not identify any dataset suitable for studying the language-incremental-domain-incremental (LIDIL) setting. Indeed, the lack of comprehensive settings that integrate both language and domain shifts, realistic episodic designs, and evolving benchmarks in existing datasets motivates our work. [Table** 3](https://anonmyous.objectstore.e2enetworks.net/nirantar-related-datasets.png) (also included in the appendix of the revised manuscript) provides a comparative overview of relevant datasets for LIL, DIL, and LIDIL scenarios. It should be noted that from this table that ours is the only dataset which contains audio samples collected from the field which are manually transcribed and have natural episodes. Our work advances CL for ASR in the following ways:
> - **A novel LIDIL setting**: Nirantar facilitates the combined study of language and domain shifts through a language-incremental-domain-incremental (LIDIL) setting. This interplay has not been explored in prior works, adding a new dimension to continual learning research.
> - **Data episodes designed from Real-World use-cases of CL**: Unlike prior works that rely on episodes crafted synthetically from existing datasets, the data episodes in Nirantar reflect real-world domain and linguistic variability.. This better represents the diversity encountered in practical applications.
> - **Longer Episodic Sequences**: Prior works are limited to shorter episodic sequences (fewer than 4 episodes for DIL and fewer than 10 for TIL). In contrast, Nirantar enables experiments over 11 episodes across all three scenarios (LIL, DIL, and LIDIL), providing a more robust comparison of CL approaches for extended learning scenarios.
> - **An Evolving Evaluation Benchmark**: Nirantar introduces an evolving benchmark by continuously adding 15-minute samples to the test set as new data is collected. Additionally, since the test data is sampled at the district level, it naturally supports evaluation in an episodic setting.
>
> **W1 (partial): “including initial statistics per language, domain, and episode”**
>
> [Table** 4](https://anonmyous.objectstore.e2enetworks.net/nirantar-dataset-detailed.png) (also included in the appendix of the revised manuscript)  presents the detailed statistics of data for each language, domain as well as episode information across all three scenarios (LIL, DIL, and LIDIL).
>
> **Q1 (Partial): “quality assurance procedures”**
>
> The data was collected and transcribed using a two-level process, following [1], involving a "maker" and a "checker." Initially, the collected data underwent a QA verification check against 20 relevant criteria, including factors such as low volume, background noise, poor extempore quality, mispronunciations, irrelevant responses and so on. Additionally, the audio was cross-verified with metadata such as gender and age group. A detailed list of these parameters can be found here [1]. To ensure transcription quality, we follow a maker-checker-superchecker workflow. After the initial transcription by the maker, the data goes through two rounds of review (checker and a superchecker). The super-checkers, who are very experienced language experts, would either instruct the checkers to rework the transcription if it was inaccurate or directly make corrections themselves, ensuring quality. With data collection and transcription carefully conducted by humans and validated through multiple levels of checks, we are confident that the likelihood of significant errors is minimal.
>
> **Q1 (Partial): “...any pre-CL performance metrics…”**
>
> We have added the pre-CL performance metrics, such as initial Word Error Rate (WER) using language specific trained models, in [Table** 4](https://anonmyous.objectstore.e2enetworks.net/nirantar-dataset-detailed.png), which is also included in the appendix of the revised manuscript
>
>
>
> [1] Javed, T., Nawale, J. A., George, E. I., Joshi, S., Bhogale, K. S., Mehendale, D., ... & Khapra, M. M. (2024). IndicVoices: Towards building an Inclusive Multilingual Speech Dataset for Indian Languages. arXiv preprint arXiv:2403.01926.

---

> ### Author Response · Authors · 2024-11-22
> **Response to reviewer YNAF (continue)**
>
> **W2: “diversity in different episodes”**
>
> Thank you for sharing your feedback. We have added [Figures** 14 to 15](https://anonmyous.objectstore.e2enetworks.net/nirantar-domain-vocab-evolution.png) (also included in the appendix of the revised manuscript), which illustrate how the domains and vocabulary evolve over episodes. From the figures, we observe a steady increase in both vocabulary count and domain coverage as we move from episode 0 to episode 11. The vocabulary and domain sizes grow consistently across all scenarios (LIL, DIL, and LIDIL). Furthermore, the change is more gradual in the LIDIL scenario compared to DIL and LIL, making it a unique scenario to study.

---

### Official Review · Reviewer_oRB3 · 2024-11-04

**Soundness:** 2
**Presentation:** 2
**Contribution:** 2
**Rating:** 5
**Confidence:** 4

**Summary:**

This paper introduces Nirantar, a dataset designed for studying continual learning methods.
The dataset comprises batches of data for various Indian languages and Indian districts.
Various continual learning methods are analyzed on the language incremental, domain incremental and language incremental domain incremental setups.

**Strengths:**

* Continual learning is an important (as it relates to data efficient learning) and unsolved problem.
* The dataset seems valuable for studying continual learning and also beyond that as it supplements existing multilingual ASR corpora with mid/high/low resource data from Indian languages.
* Experimentation is reasonably thorough and tests different CL methods in various setups.

**Weaknesses:**

* It's unclear whether the design of the dataset makes it unique to test CL methods. For example, a similar scheme could be adopted by taking the various releases over time from Common Voice (the domains would not be obtained in a straightforward manner though). The paper does not really demonstrate that the dataset provides value over  taking an existing dataset and slicing it into episodes.
* Related to the previous point, if the methodology to collect the data is the same as for IndicVoices, the contribution for this paper becomes less clear (other than adding more volume).
* A more minor weakness: the writing could be improved, for example by justifying all empirical choices (see clarification questions)

**Questions:**

* 269: “we select the 11 languages having the largest number of hours in Table 1”. Why is the largest number of hours chosen?
* 309: “We create timelines of length τ = 11”. Why was 11 chosen?
* 377: “Average MER: Match Error Rate”. Why not use WER or CER?

presentation: the citation formatting is unusual and can be double checked/fixed.

---

> ### Author Response · Authors · 2024-11-22
> **Response to reviewer oRB3**
>
> We sincerely thank the reviewer for their useful feedback, please find our responses below.
>
>     Items marked with ** indicate hyperlinks
>
> **W1 & W2: “design of datasets, volume of data”**
>
> While we acknowledge the existence of certain datasets that can be repurposed for continual learning in LIL or DIL scenarios, we could not identify any dataset suitable for studying the language-incremental-domain-incremental (LIDIL) setting. Indeed, the lack of comprehensive settings that integrate both language and domain shifts, realistic episodic designs, and evolving benchmarks in existing datasets motivates our work. [Table** 3](https://anonmyous.objectstore.e2enetworks.net/nirantar-related-datasets.png) (also included in the appendix of the revised manuscript) provides a comparative overview of relevant datasets for LIL, DIL, and LIDIL scenarios. It should be noted that from this table that ours is the only dataset which contains audio samples collected from the field which are manually transcribed and have natural episodes. Our work advances CL for ASR in the following ways:
> - **A novel LIDIL setting**: Nirantar facilitates the combined study of language and domain shifts through a language-incremental-domain-incremental (LIDIL) setting. This interplay has not been explored in prior works, adding a new dimension to continual learning research.
> - **Data episodes designed from Real-World use-cases of CL**: Unlike prior works that rely on episodes crafted synthetically from existing datasets, the data episodes in Nirantar reflect real-world domain and linguistic variability.. This better represents the diversity encountered in practical applications.
> - **Longer Episodic Sequences**: Prior works are limited to shorter episodic sequences (fewer than 4 episodes for DIL and fewer than 10 for TIL). In contrast, Nirantar enables experiments over 11 episodes across all three scenarios (LIL, DIL, and LIDIL), providing a more robust comparison of CL approaches for extended learning scenarios.
> - **An Evolving Evaluation Benchmark**: Nirantar introduces an evolving benchmark by continuously adding 15-minute samples to the test set as new data is collected. Additionally, since the test data is sampled at the district level, it naturally supports evaluation in an episodic setting.
>
> **W3 & Q1, Q2, Q3: “Empirical justification”**
>
> - The rationale behind using continual learning (CL) is to adapt a good base model to perform well on fresh data, which may come in the form of new languages, new domains, or both. Therefore, it is critical to start with a robust base model to test CL approaches effectively. For this reason, the largest number of hours was chosen, as it ensures sufficient data for establishing a solid foundation in the initial model. In our experiments, we make the following choices across scenarios:
> LID: Start with half the languages as the base model.
> DIL: Start with half the districts for each language as the base model.
> LIDIL: Start with half the languages with half the districts as the base model.
> - The remaining of the episodes corresponds to the remaining number of languages (11) in our setup. However, this does not limit the episodes for DIL and LIDIL to a maximum of 11. Our evaluation framework is designed to evolve over time, enabling the addition of new episodes for both DIL and LIDIL scenarios.
> - WER and CER can exceed 100%, leading to inconsistencies when comparing different approaches, particularly in high-error scenarios. To ensure a more standardized and interpretable evaluation, we use Match Error Rate (MER) as the primary metric. MER is bounded within the range [0, 1], where 0 represents no error between the ground truth and the model hypothesis, and 1 represents a completely incorrect hypothesis. For completeness, we will include WER and CER plots in the appendix for all our results.
> - We apologize for the citation style, we have updated the manuscript with the revised citation style.

---

> > ### Comment · Reviewer_oRB3 · 2024-11-26
> >
> > Thank you for the very detailed answer addressing my comments.
> > I may have missed this in the rebuttal but could you clarify what novel aspects Nirantar brings beyond IndicVoices, besides increased data volume? For example, are there any differences in data collection, organization, or intended use cases that distinguish Nirantar?

---

> > > ### Author Response · Authors · 2024-11-26
> > > **Response to Reviewer oRB3**
> > >
> > > We would like to clarify the distinct contributions of Nirantar, emphasizing that its novelty goes far beyond simply increasing the volume of data relative to IndicVoices. While data volume is one axis of comparison, Nirantar introduces multiple new dimensions that were not within the scope of IndicVoices. Below, we outline the key differences:
> > >
> > > - **Distinct Purpose and Focus on Continual Learning (CL):** IndicVoices was designed as a foundational dataset for building multilingual ASR systems and was never intended to address the challenges of continual learning. Its primary goal was to establish a comprehensive dataset for static model training. Recognizing the practical challenge of training downstream models with incremental data, Nirantar explicitly incorporates a continual learning (CL) framework as one of its core goals. This focus fundamentally shifts the purpose of the dataset from static ASR development to exploring the complexities of training models incrementally with multilingual and multi-domain data inflow.
> > >
> > > - **Beyond Dataset Introduction: A Novel CL Framework:** Nirantar does not simply introduce a dataset but establishes a new lens for research in CL. It supports three Incremental Learning (IL) scenarios: Language-Incremental (LIL), Domain-Incremental (DIL), and the novel Language-Incremental Domain-Incremental Learning (LIDIL). LIDIL, in particular, has never been studied in the literature, making this work a first-of-its-kind contribution to CL research. This dimension of the work was entirely absent in IndicVoices, which did not address CL scenarios at all.
> > >
> > > - **Curated Test Sets for Continual Evaluation:** Nirantar introduces carefully curated test sets designed for continual evaluation. These test sets ensure that models trained incrementally can be assessed meaningfully, both during development and in future work. In contrast, IndicVoices provides a static test set (5 hours per language) that is well-suited for general-purpose ASR evaluation but entirely unsuitable for studying CL scenarios. This distinction highlights the forward-looking intent of Nirantar to enable robust continual evaluation over time.
> > >
> > > - **Data Collection and Scale:** While the collection process for Nirantar builds upon the methodology introduced in IndicVoices, achieving the increased scale required significant additional effort. The sheer logistical and operational complexity of adding this volume should not be discounted, even if the same underlying collection process was followed. Scaling to such an extent, while maintaining quality and diversity, required a substantial effort.
> > >
> > > **Summary**
> > >
> > > While the collection process overlaps with IndicVoices, Nirantar represents a fresh look at the problem by addressing entirely new challenges and enabling research that was not in the intended scope of IndicVoices. Its focus on continual learning, novel IL scenarios, and tailored evaluation frameworks firmly establish Nirantar as a distinct and substantial contribution, not merely an incremental addition to IndicVoices.

---

> > > > ### Comment · Reviewer_oRB3 · 2024-12-03
> > > >
> > > > Thank you for the additional explanation about the contribution beyond increasing the volume of IndicVoices. I will raise my score accordingly.

---

### Official Review · Reviewer_XYQq · 2024-11-04

**Soundness:** 3
**Presentation:** 3
**Contribution:** 3
**Rating:** 6
**Confidence:** 4

**Summary:**

This paper introduces Nirantar, a dataset designed to facilitate continual learning (CL) through real-world, large-scale speech data from 22 Indian languages. Data is gathered incrementally across diverse languages and domains, making it distinct from typical CL datasets that rely on simulated episodes.

Nirantar support 3 key CL scenarios: Language-Incremental (LIL), Domain-Incremental (DIL), and the novel Language-Incremental Domain-Incremental Learning (LIDIL), which has not previously been studied. The authors’ evaluation of several existing CL algorithms in these scenarios reveals that algorithmic behavior varies significantly across them, suggesting the need for dedicated analyses tailored to each scenario.

**Strengths:**

The dataset’s design closely mirrors real-world CL requirements, providing a valuable resource for research on multilingual, multi-domain, and incremental learning.

The inclusion of both widely used CL scenarios (LIL, DIL) and the novel LIDIL scenario represents a significant contribution, opening new avenues for research.

Evaluations of CL approaches provide useful initial insights, underscoring the complexity and variability of performance across the dataset’s scenarios.

**Weaknesses:**

Figures 3-5 in the report show fluctuating trends, which might indicate an imbalance in the amount of data available for each language. This imbalance could impact the model's performance on specific low-resource languages, potentially limiting its effectiveness in some areas.


While the evaluations provide a starting point, a deeper exploration of algorithmic adaptations or optimizations specifically for LIDIL could enhance the study’s practical impact.

**Questions:**

Given the natural variations in data size per language and domain, what strategies are recommended to handle data imbalance in the CL models?

---

> ### Author Response · Authors · 2024-11-22
> **Response to reviewer XYQq**
>
> We sincerely thank the reviewer for their useful feedback, please find our responses below.
>
> **W1: “fluctuating trends”**
>
> The fluctuations observed in Figures 3-5 are not solely due to data imbalance across languages but rather highlight the need for improved continual learning (CL) approaches to handle episodic data effectively. For instance, in the LIL scenario, the fluctuations in Forward Transfer (FT) and Intransigence reflect that certain data batches are inherently more challenging.
> A specific example can be seen when transitioning from episode 8 to episode 9 in the LIL scenario, where Manipuri—a Tibeto-Burman language that is significantly different from the previously seen languages—is introduced. This results in a sharp decline in FT, as the model struggles to adapt to this distinct language while retaining information from prior tasks. We view this as a strength of our dataset, as it provides valuable insights into the challenges faced by CL methods and helps evaluate them more effectively.
>
> **Q1: “strategies for data imbalance”**
>
> In continual learning (CL) models, data imbalance across languages and domains can lead to biased learning, where the model favours more frequently encountered classes, thus diminishing performance on underrepresented categories. In our study, we use temperature-based sampling [1,2] which adjusts the probability of selecting samples based on their rarity in the episode. This is a widely used method when training multilingual models [3] where data imbalance across languages is unavoidable. In the context of data imbalance, it can help ensure that minority class examples are more likely to be chosen for training. By increasing the "temperature" in the sampling process, the model reduces bias toward more frequently occurring data points and encourages more balanced exposure to various classes​. This is particularly important when training with episodes involving multiple languages (DIL and LIDIL setting) and does not apply for LIL where a single language is introduced at each time. In our experiments we set alpha to 1.5
> In addition to this, data augmentation techniques, such as synthetic data generation [4], are widely used to address class imbalance. These methods artificially increase the representation of underrepresented classes by generating additional data, thereby balancing their occurrence with other classes in the dataset. We have not explored the use of synthetic data as it is beyond the scope of our current work.
>
> [1] Aharoni, R., Johnson, M., & Firat, O. (2019, June). Massively Multilingual Neural Machine Translation. In Proceedings of the 2019 Conference of the North American Chapter of the Association for Computational Linguistics: Human Language Technologies, Volume 1 (Long and Short Papers) (pp. 3874-3884).
>
> [2] Wang, X., Tsvetkov, Y., & Neubig, G. (2020, July). Balancing Training for Multilingual Neural Machine Translation. In Proceedings of the 58th Annual Meeting of the Association for Computational Linguistics (pp. 8526-8537).
>
> [3] Conneau, A., & Lample, G. (2019). Cross-lingual language model pretraining. Advances in neural information processing systems, 32.
>
> [4] Shorten, C., Khoshgoftaar, T.M. A survey on Image Data Augmentation for Deep Learning. J Big Data 6, 60 (2019). https://doi.org/10.1186/s40537-019-0197-0
>
>
> [4] Shorten, Connor and Taghi M. Khoshgoftaar. “A survey on Image Data Augmentation for Deep Learning.” Journal of Big Data 6 (2019): 1-48.

---

### Official Review · Reviewer_JqUP · 2024-11-04

**Soundness:** 3
**Presentation:** 4
**Contribution:** 4
**Rating:** 8
**Confidence:** 4

**Summary:**

The paper introduces Nirantar, a new benchmark for continual learning (CL) in automatic speech recognition (ASR) that deals with real-world challenges of adding new languages and domains over time. Nirantar is based on a large dataset of 3,250 hours of speech across 22 Indian languages and 208 districts (domains), collected in phases that mimic real-world, incremental data collection.

The main contributions of the paper are:

1) New Data: The paper relies on existing dataset, but contributes an additional 1720 hours of newly collected speech data.

2) Real Scenarios: the benchmark provides realistic settings for continual learning, including Language-Incremental Learning (LIL), Domain-Incremental Learning (DIL), and a novel Language-Incremental Domain-Incremental Learning (LIDIL), explored in ASR for the first time.

3) Comprehensive Benchmarking: The paper evaluates several popular CL methods (e.g., Experience Replay, Elastic Weight Consolidation) across all scenarios, offering a thorough comparison of CL approaches.

4) Open-Source Contributions: All resources from this work will be made publicly available to encourage further research in multilingual and multi-domain continual learning for ASR.

In summary, Nirantar provides a valuable benchmark that reflects real-world data collection and introduces new challenges for continual learning in speech recognition.

**Strengths:**

- **Novel Contribution:** The work is original, not only contributing newly collected data, but including a novel Continual Scenario for ASR: Language-Incremental Domain-Incremental Learning (LIDIL). This scenario provides a more realistic view of real-world multilingual and multi-domain environments compared to many CL datasets that rely on synthetic or structured data, making its impact relevant beyond the specific languages covered.
Comprehensive Evaluation: The evaluation is extensive, covering popular continual learning methods like Experience Replay, Elastic Weight Consolidation, and Memory-Aware Synapse, along with meaningful metrics. The benchmark also evaluates CL methods across three distinct scenarios (LIL, DIL, and LIDIL), offering a detailed and multi-faceted analysis that allows for a nuanced understanding of model behavior in varying incremental learning conditions.

- **Clarity and Structure:** The paper is structured logically, with clear definitions and explanations of the continual learning scenarios and metrics. The description of data collection, experimental setups, and training procedures is transparent, with steps and design choices explained in detail. This clarity makes it easier for researchers to replicate or build upon the work.

- **High Impact for Low-Resource ASR:** The benchmark is valuable for advancing low-resource ASR research, especially for underrepresented languages like Indian languages. By focusing on multilingual, multi-domain continual learning, the benchmark tackles unique challenges in low-resource ASR.

- **Open Access:** the open release of the dataset and resources under a permissive license further promotes research and collaboration in the field, supporting wider accessibility of multilingual ASR technology.

**Weaknesses:**

- **Limited Baselines:** While the paper evaluates CL methods from replay-based and regularization-based approaches, it excludes architecture-based methods arguing they are impractical for real-world settings as they add parameters for each new language and domain, leading to excessive complexity as episodes increase (lines 328-332). However, this overlooks recent advances in lightweight architecture-based methods like language-specific adapters, which have improved ASR performance even in domains harder than the training data [1]. Including such methods could provide valuable baselines and insights into scalable alternatives that balance parameter efficiency and adaptability.

- **Discussions on Cross-Lingual Transfer:** The dataset is highly diverse, covering 22 Indian languages and 208 districts, but this diversity is underexplored in the analysis. A more granular breakdown of metrics would allow a clearer view of knowledge transferability across closely related languages and domains. For instance, including subgroup analyses by geographic distance, language family, or dialectal variations could highlight the impact of linguistic and regional proximity on model performance. Such details would enhance understanding of the effectiveness of CL techniques in low-resource settings. For example, on line 465, the authors attribute a performance decline to the introduction of a Tibeto-Burman language into the dataset predominantly composed of Indo-Aryan and Dravidian languages. Expanding on this, with metrics grouped by language family or linguistic characteristics, would provide more actionable insights into cross-language transfer. Additionally, including dataset attributes like vocabulary overlap, phonetic variations, or language family classification would better showcase the dataset’s uniqueness and clarify the transferability of findings to other low-resource multilingual datasets.

References

[1] Ferraz, T. P., Boito, M. Z., Brun, C., & Nikoulina, V. (2024). Multilingual Distilwhisper: Efficient Distillation of Multi-Task Speech Models Via Language-Specific Experts. ICASSP 2024.

**Questions:**

1) Could you provide details on how human subjects were compensated and whether explicit consent was obtained for both research and potential industrial use of their voices? Including protocols and consent guidelines in the appendix would be helpful.

2) I recommend providing sample data in the appendix to illustrate the dataset.

**Details Of Ethics Concerns:**

The authors provide information and assert that these issues have been address; however, I am not entirely sure this is sufficient for acceptance. Therefore, I recommend an ethical review that considers these points:
- **Consent and Data Usage:** Details on whether participants provided informed consent specifically for both research and potential industrial applications of the data, and how data anonymization was ensured to protect privacy.
- **Fair Compensation:** Discussion on whether compensation for data contributors and annotators aligns with fair local standards to prevent exploitation.

---

> ### Author Response · Authors · 2024-11-22
> **Response to reviewer JqUP**
>
> We sincerely thank the reviewer for their useful feedback, please find our responses below.
>
>     Items marked with ** indicate hyperlinks
>
> **W1: “Limited baselines, architecture based methods”**
>
> As discussed in Section 1 of the paper, in continual learning (CL) settings involving a large number of languages and domains, architecture-based approaches can lead to model bloat and unnecessary complexity. However, based on the reviewer’s suggestion, we explored this approach in the LIL scenario, where we added up to 11 adapters  (one for every new language). These adapters were integrated into each Conformer block of the Conformer-L model, with a bottleneck dimension of 64, resulting in an additional 1 million parameters per language. The results of this experiment are presented in [Figure** 10](https://anonmyous.objectstore.e2enetworks.net/nirantar-lil-adapters.png) (also included in the appendix of the revised manuscript):
> - The Adapters method outperforms all other CL approaches, except for ER, in terms of AMER and Backward Transfer. This is primarily because each new episode adds an adapter layer, which prevents forgetting during the training process, as each episode trains a different adapter without modifying the base model. Interestingly, the difference between Joint FT and Adapters can be attributed to the number of parameters involved. We believe that increasing the adapter's bottleneck dimension to expose more trainable parameters could further reduce the gap.
> - Forward transfer is worse for adapters because they are specifically tuned for individual languages in each episode, without facilitating knowledge transfer to future episodes. This limits the ability to leverage shared knowledge across languages and domains, which could benefit subsequent tasks.
> - Adapters exhibit the highest Intransigence Measure, as the entire backbone stays frozen, and only the language-specific adapters are updated during each episode. This introduces rigidity, but it also helps mitigate catastrophic forgetting. That said, during this experiment, the number of parameters increased by 11 million (1 million per episode). If extended to a domain-incremental or language-incremental-domain-incremental setting, the parameter count could reach an order of magnitude of O(100), making it impractical for real-world applications.
>
> We will add the above results to the final version of the paper and highlight that ER based approaches still remain a more feasible alternative as it performs better on most metrics and is applicable in all the three settings (LIL, DIL, LIDIL). Thank you again for suggesting this experiment as we believe it has helped make our analysis more comprehensive.
>
> **W2: “Cross-lingual transfer”**
>
> We study the cross-lingual transfer of information for two language families, Indo-Aryan and Dravidian, in the LIDIL setting. [Figures** 11 to 13](https://anonmyous.objectstore.e2enetworks.net/nirantar-crosslingual-lidil.png) (also included in the appendix of the revised manuscript) illustrates the joint results (row 1) as well as results spliced by language families (row 2 and row 3). It can be clearly seen that the Average MER (AMER) and Backward Transfer (BT) for the Dravidian languages are better than for Indo-Aryan languages. This could imply that Dravidian languages are more related to each other than Indo-Aryan languages, facilitating better transfer of information. Additionally, the smaller number of languages in the Dravidian group could also contribute to these improved results, as fewer languages may reduce the complexity and help the model focus on the shared linguistic features within this group. The overall trends, however, suggest that Dravidian languages show more stable learning behavior in the LIDIL setting compared to Indo-Aryan languages.
>
> We thank the reviewer for suggesting the possibility of exploring crosslingual transfer. We will add the above analysis and results to the final version of the paper and hope that this helps in addressing this important question.

---

> ### Author Response · Authors · 2024-11-22
> **Response to reviewer JqUP (continue)**
>
> **Q1 & Q2: "Compensation, consent, guidelines"**
>
> We will include additional details (participant instructions mentioned below) and sample data examples in the appendix for clarity.
>
> Referring to Section 7, we would like to highlight that the data collection process adheres to the guidelines established for IndicVoices [1]. This process was thoroughly reviewed and approved by the Institute Ethics Committee. Participants were fully informed about the data collection, their involvement, and the use of their data, and their consent was obtained beforehand. They received compensation aligned with local daily wages for their time and effort. No PII data will be shared externally, and measures were implemented to anonymize and protect sensitive information. Project staff were also compensated appropriately. Nirantar will be released under the CC-BY-4.0 license, permitting commercial use.
> The following refers to the instructions shared to the Participants
> - **Aim of the Project**: The project aims to collect data for developing and evaluating speech technology tailored to your language. This technology includes everyday applications explained by the coordinator.
> - **Purpose of Data Collection**: Your data will be used for both commercial and non-commercial purposes to improve speech technology for your language.
> - **Amount of Time**: The process will take approximately 1 to 4 hours.
> - **Compensation**: You will be compensated with INR X, as communicated by the coordinator. This amount X is equivalent to the local half-day wage in the region.
> - **Consent**: By participating, you agree to the terms outlined, including the use of your data as described. Please proceed only if you consent and sign the consent form.
> - **Registration**: Your details will be collected during registration. To ensure privacy, none of this information will be shared with third parties. You are required to upload your signed consent form as part of the registration process.
> - **App Installation**: Download the “<Anonymous>” app from the Google Play Store on your Android device.
> - **Login**: Log in using the access code assigned to you at registration. Verify this code with the coordinator.
> - **Fetch Tasks**: Click “Submit Tasks/Get New Tasks” to view your assigned tasks.
> Recording: Click on each task, read the prompt, and record your response. If unsure about a task, ask the coordinator for clarification. Take breaks as needed and complete tasks at your own pace. You may skip questions if uncomfortable.
> - **Submit a Task**: After recording, click “Stop” to replay your response. If you and/or the coordinator are satisfied, click "Next". If not, re-record.
> - **Two-party Conversations**: After completing app tasks, you’ll be paired with another participant for a scenario-based conversation. You can select your partner, role, and scenario from available options shared by the coordinator.
> - **Logging Out**: Once all tasks are complete, return to the home screen and click “Submit Tasks”. Wait for confirmation, then record a video for identity verification.
>
> [1] Javed, T., Nawale, J. A., George, E. I., Joshi, S., Bhogale, K. S., Mehendale, D., ... & Khapra, M. M. (2024). IndicVoices: Towards building an Inclusive Multilingual Speech Dataset for Indian Languages. arXiv preprint arXiv:2403.01926.

---

> > ### Comment · Reviewer_JqUP · 2024-11-26
> >
> > Thank you for your detailed and thoughtful responses. I also took the time to read the comments and responses provided to other reviewers.
> >
> > Regarding the issues of compensation and user consent, I am personally satisfied with the steps taken to address these concerns. However, I defer to the Ethics Reviewer for the final decision, as I don't have sufficient experience in this area and cannot assess whether the measures meet the conference's thresholds.
> >
> > I appreciate the additional experiments with adapters and the expanded discussion on cross-lingual transfer, which significantly enhance the analysis and strengthen the paper's contributions.
> >
> > I maintain my original score, I believe the paper should be accepted.

---

### Official Review · Reviewer_6Bxs · 2024-11-04

**Soundness:** 3
**Presentation:** 3
**Contribution:** 2
**Rating:** 5
**Confidence:** 4

**Summary:**

Nirantar is a large-scale dataset featuring 3,250 hours of human-transcribed speech across 22 languages from 208 districts in India, with 1,720 hours newly released in this work. Collected in incremental batches introducing new languages and locations, Nirantar creates a real-world continual learning (CL) setting that supports training and evaluation in Language-Incremental (LIL), Domain-Incremental (DIL), and the novel Language-Incremental Domain-Incremental Learning (LIDIL) scenarios. Initial evaluations show that CL approaches behave differently across these scenarios, highlighting the dataset's utility for diverse CL research.

**Strengths:**

- The dataset encompasses a diverse range of languages across India.
- They offer three distinct continual learning settings: Language-Incremental (LIL), Domain-Incremental (DIL), and the novel Language-Incremental Domain-Incremental Learning (LIDIL).
- The paper is well-written and easy to follow, with a comprehensive analysis included.

**Weaknesses:**

- The first two settings, Language-Incremental and Domain-Incremental, have been explored in previous studies, including CL-mARS, which covers various CL methods in multilingual contexts (https://arxiv.org/pdf/2310.16931), and works by Sadhu et al. (https://www.isca-archive.org/interspeech_2020/sadhu20_interspeech.pdf), Chang et al. (https://arxiv.org/abs/2104.01616), and Li et al. (https://arxiv.org/abs/2302.01496), which investigate incremental domain setups. In light of these prior studies, the contribution here feels mostly incremental in these areas.

- The dataset focuses exclusively on languages from India, which provides rich diversity, though it could be beneficial to include an analysis of language relationships and similarities to understand their potential impact on performance. Expanding the dataset with additional languages from public datasets like FLEURS or CommonVoice, especially those with minimal similarity, would further strengthen the study.

- Additionally, task order is determined randomly, yet exploring the effects of task ordering on performance or providing an analysis that considers language similarity, task order, and performance metrics could add valuable insights.

- Finally, while the study examines rehearsal-based and regularization-based CL methods, incorporating architecture-based approaches, such as Progressive Neural Networks (PNN), Piggyback (PB), and Learning to Prompt (L2P), could offer a more comprehensive view, as highlighted in prior literature.

**Questions:**

Refer to weakness

**Details Of Ethics Concerns:**

Since this work introduces a new dataset, it’s essential to ensure that privacy and ethical practices are observed. Notably, Section 7 outlines the ethical guidelines and practices they follow.

---

> ### Author Response · Authors · 2024-11-22
> **Response to reviewer 6Bxs**
>
> We sincerely thank the reviewer for their useful feedback, please find our responses below.
>
>     Items marked with ** indicate hyperlinks
>
> **W1: “... contribution here feels mostly incremental …”**
>
> We have cited the prior works, including CL-MASR, Sadhu et al., Chang et al., and Li et al., and duly acknowledged their contributions in the related work section. Indeed, the lack of comprehensive settings that integrate both language and domain shifts, realistic episodic designs, and evolving benchmarks in existing datasets motivates our work. Specifically, in contrast to these works, our work makes the following contributions:
> - **A novel LIDIL setting**: Nirantar facilitates the combined study of language and domain shifts through a language-incremental-domain-incremental (LIDIL) setting. This interplay has not been explored in prior works, adding a new dimension to continual learning research.
> - **Data episodes designed from Real-World use-cases of CL**: Unlike prior works that rely on episodes crafted synthetically from existing datasets, the data episodes in Nirantar reflect real-world domain and linguistic variability.. This better represents the diversity encountered in practical applications.
> - **Longer Episodic Sequences**: Prior works are limited to shorter episodic sequences (fewer than 4 episodes for DIL and fewer than 10 for TIL). In contrast, Nirantar enables experiments over 11 episodes across all three scenarios (LIL, DIL, and LIDIL), providing a more robust comparison of CL approaches for extended learning scenarios.
> - **An Evolving Evaluation Benchmark**: Nirantar introduces an evolving benchmark by continuously adding 15-minute samples to the test set as new data is collected. Additionally, since the test data is sampled at the district level, it naturally supports evaluation in an episodic setting.
>
> Furthermore, [Table** 3](https://anonmyous.objectstore.e2enetworks.net/nirantar-related-datasets.png) (also included in the appendix of the revised manuscript) presents a comparative overview of relevant datasets that can be used in LIL, DIL and LIDIL scenarios. It should be noted that from this table, ours is the only dataset which contains audio samples collected from the field which are manually transcribed and have natural episodes.
>
> **W2: “... Expanding the dataset with additional languages from public datasets …”**
>
> We agree that adding more languages from the public datasets and analyzing language relationships and similarities could provide additional insights. However, we would like to highlight some limitations for the suggested datasets:
> - While FLEURS provides linguistic diversity with data for 102 languages, it primarily consists of read speech, derived from the dev and devtest splits of the FLoRes dataset. This design imposes significant constraints on its utility for studying CL paradigms:
>     - It lacks domain information to enable domain-incremental and language-incremental-domain-incremental learning.
>     - Due to its design, the dataset contains 10 hours of training data per language, which is not sufficient to use it even in the language-incremental learning setting.
> - Although Common Voice is large and diverse, consisting of data for 131 languages, it lacks domain information, making it unsuitable for studying domain-incremental learning (DIL) or the language-incremental-domain-incremental (LIDIL) setting.
>
> We have also reviewed other existing publicly available datasets, but none meet the requirements for evaluating CL models across all three scenarios (LIL, DIL, and LIDIL). We once again point to [Table** 3](https://anonmyous.objectstore.e2enetworks.net/nirantar-related-datasets.png) (also included in the appendix of the revised manuscript) which presents a comparative overview of relevant datasets that can be used in LIL, DIL and LIDIL scenarios.

---

> ### Author Response · Authors · 2024-11-22
> **Response to reviewer 6Bxs (continue)**
>
> **W3: “... task order is determined randomly …”**
>
> We acknowledge that the task order was determined randomly in the paper. To resolve this, we performed the study with two more random orderings to highlight that our observations and analysis stay consistent with other random orderings as well. The following lines list the original task order and two more permutations of it for the LIDIL scenario.
>
>     Random Order 1	: 0→1→2→3→4→5→6→7→8→9→10→11
>     Random Order 2	: 0→11→1→2→10→8→5→9→3→4→6→7
>     Random Order 3  : 0→8→6→7→9→4→5→1→2→3→11→10
>
> [Figures** 7 to 9](https://anonmyous.objectstore.e2enetworks.net/nirantar-permute-lidil.png) (also included in the appendix of the revised manuscript) present the results for the original episodic sequence (Random Order 1) and two additional randomized sequences (Random Order 2 and Random Order 3) in the LIDIL scenario.
> - **Consistency Across Permutations**: The AMER and BT performance of different continual learning methods (Inc. FT, ER, EWC, MAS) remains similar regardless of the episode order.
> - **Stable Performance Rankings**: The relative rankings of methods (in terms of Forward Transfer, Backward Transfer, and Intransigence) are similar across all permutations. This shows that the order of episodes doesn’t drastically affect how methods compare in terms of learning new tasks and preserving previous knowledge.
> - **Impact of Episode Order on Intransigence**: Intransigence fluctuates with episode order, suggesting that certain episode sequences are more challenging to train. For instance, episode 2 in the original order appears as episode 3 and episode 8 in the permuted orders (1 and 2), which corresponds to noticeable peaks in the intransigence measure.
>
> **W4: “... architecture-based approaches…”**
>
> As discussed in Section 1 of the paper, in continual learning (CL) settings involving a large number of languages and domains, architecture-based approaches can lead to model bloat and unnecessary complexity. However, based on the reviewer’s suggestion, we explored this approach in the LIL scenario, where we added up to 11 adapters  (one for every new language). These adapters were integrated into each Conformer block of the Conformer-L model, with a bottleneck dimension of 64, resulting in an additional 1 million parameters per language. The results of this experiment are presented in [Figure** 10](https://anonmyous.objectstore.e2enetworks.net/nirantar-lil-adapters.png) (also included in the appendix of the revised manuscript):
> - The Adapters method outperforms all other CL approaches, except for ER, in terms of AMER and Backward Transfer. This is primarily because each new episode adds an adapter layer, which prevents forgetting during the training process, as each episode trains a different adapter without modifying the base model. Interestingly, the difference between Joint FT and Adapters can be attributed to the number of parameters involved. We believe that increasing the adapter's bottleneck dimension to expose more trainable parameters could further reduce the gap.
> - Forward transfer is worse for adapters because they are specifically tuned for individual languages in each episode, without facilitating knowledge transfer to future episodes. This limits the ability to leverage shared knowledge across languages and domains, which could benefit subsequent tasks.
> - Adapters exhibit the highest Intransigence Measure, as the entire backbone stays frozen, and only the language-specific adapters are updated during each episode. This introduces rigidity, but it also helps mitigate catastrophic forgetting. That said, during this experiment, the number of parameters increased by 11 million (1 million per episode). If extended to a domain-incremental or language-incremental-domain-incremental setting, the parameter count could reach an order of magnitude of O(100), making it impractical for real-world applications.
>
> We will add the above results to the final version of the paper and highlight that ER based approaches still remain a more feasible alternative as it performs better on most metrics and is applicable in all the three settings (LIL, DIL, LIDIL). Thank you again for suggesting this experiment as we believe it has helped make our analysis more comprehensive.

---

### Author Response · Authors · 2024-11-30
**Gentle reminder to all reviewers**

We would like to sincerely thank all the reviewers for their thoughtful feedback and constructive suggestions. We have made every effort to address the comments and improve the manuscript. We are also grateful to reviewers **JqUP** and **oRB3** for their prompt replies to our initial comments. As the deadline for the author-reviewer discussion phase (**Dec 2nd**) is approaching, we kindly request reviewers **6Bxs**, **XYQq**, **YNAF**, and **oRB3** to check our responses and let us know if any further clarifications are needed from our side. If all the concerns are addressed we request the reviewers to appropriately increase their scores.

---

### Author Response · Authors · 2024-12-03
**Rebuttal Summary**

We sincerely appreciate the reviewers for their insightful and constructive feedback. Their suggestions, including exploring different task orderings and integrating adapters, have been instrumental in enhancing the quality and depth of our manuscript. Below, we present a summary of the key points addressed in our rebuttal, as highlighted by the reviewers:

 - **(6Bxs, YNAF)**: Highlighted limitations of existing datasets (e.g., FLEURS, Common Voice, GigaSpeech 2) for studying DIL and LIDIL scenarios as they lack domain information. Presented a comparative analysis, showing Nirantar as the only dataset with real-world, manually transcribed, and episodic data designed for Continual learning (CL). [[Table** 3]](https://anonmyous.objectstore.e2enetworks.net/nirantar-related-datasets.png)

 - **(6Bxs)**: Addressed randomness in task order by including results for two additional permutations for the LIDIL scenario, showing consistency across different orderings. [[Figures** 7 to 9]](https://anonmyous.objectstore.e2enetworks.net/nirantar-permute-lidil.png)

 - **(6Bxs, JqUP)**: Conducted experiments with adapters in the LIL setting, showing their effectiveness in preventing catastrophic forgetting during training. However, we still caution against the possibility of parameter bloat in scenarios when a large number of adapters are added (e.g., in DIL and LIDIL scenarios) and the limited scalability of adapters compared to ER-based approaches. [[Figure** 10]](https://anonmyous.objectstore.e2enetworks.net/nirantar-lil-adapters.png)

 - **(JqUP)**: Explored cross-lingual transfer in LIDIL scenario for Indo-Aryan and Dravidian languages, revealing stable and better learning for Dravidian languages. [[Figures** 11 to 13]](https://anonmyous.objectstore.e2enetworks.net/nirantar-crosslingual-lidil.png)

 - **(JqUP)**: Clarified detailed participant instructions, consent and guidelines, which shall be duly added to the final version of the manuscript.

 - **(XYQq)**: Addressed the comment on "fluctuating trends", explaining that they reflect challenges of continual learning (CL) methods, such as adapting to significantly different new languages (e.g., Manipuri in LIL scenario) rather than mere data imbalance.

 - **(XYQq)**: Discussed strategies for addressing data imbalance.

 - **(6Bxs, oRB3)**: Highlighted Nirantar's unique contributions, including the novel LIDIL setting, real-world episodic designs, longer episodic sequences, and evolving evaluation benchmark that distinguish it from existing datasets. Emphasized that Nirantar's focus on CL, novel IL scenarios, and tailored evaluation frameworks mark it as a distinct advancement beyond IndicVoices.

 - **(oRB3)**: Justified empirical choices in LIL, DIL, and LIDIL scenarios and clarified the use of Match Error Rate (MER) over WER/CER for consistent evaluation. Additional plots for WER and CER will be included in the final manuscript.

 - **(YNAF)**: Provided statistics for language, domain, and episodes [[Table** 4]](https://anonmyous.objectstore.e2enetworks.net/nirantar-dataset-detailed.png) and demonstrated diversity in episodes through evolving vocabulary and domains. [[Figures** 14 to 15]](https://anonmyous.objectstore.e2enetworks.net/nirantar-domain-vocab-evolution.png)

 - **(YNAF)**: Added details of multi-level QA processes for data collection and transcription to ensure minimal errors and included pre-CL performance metrics using language-specific models. [[Table** 4]](https://anonmyous.objectstore.e2enetworks.net/nirantar-dataset-detailed.png)

---

### Meta-Review · Area_Chair_NB1d · 2024-12-19

**Metareview:**

The paper introduces Nirantar, a large-scale dataset and benchmark for continual learning (CL) in automatic speech recognition (ASR), focusing on real-world challenges of incremental data collection. Nirantar comprises 3,250 hours of human-transcribed speech spanning 22 Indian languages and 208 districts (domains), with 1,720 hours newly contributed in this work. The work claimed following contribution: 1) a dataset collected in a realist scenario mimicking real-world multilingual and multidoma in CL challenges, where audios were collected incrementally across languages and locations. The dataset support various CL scenarios including Language-Incremental Learning (LIL), Domain-Incremental Learning (DIL), and Language-Incremental Domain-Incremental Learning (LIDIL); 2) initial benchmarking the dataset on several existing CL methods (e.g., Experience Replay, Elastic Weight Consolidation, Memory-Aware Synapse) across these scenarios; authors also promised to make the datasets along with evaluation benchmarks publicly available


Strength of this paper
- Built a dataset with diversity in domains/languages and reflecting real-world multilingual and multi-domain continual learning (CL) challenges, supporting incremental learning and adaptation across diverse languages and domains The dataset allows exploration of a new scenario, Language-Incremental Domain-Incremental Learning (LIDIL), beyond traditional Language-Incremental (LIL) and Domain-Incremental (DIL) settings. LIDIL offers a more realistic view of multilingual, multi-domain challenges and new research problems.
- Authors also conduct benchmarking and evaluation to evaluates several established CL methods, including Experience Replay, Elastic Weight Consolidation, and Memory-Aware Synapse across various scenarios (LIL, DIL, and LIDIL), understand the characteristics of the datasets, and provide analysis and insight for future research
- The dataset and resources will be released under CC-BY-4.0 license, encouraging collaboration and broader accessibility in multilingual ASR research. The dataset will be a valuable resource for the speech research community, setting a new benchmark for multilingual, multidomain, and incremental learning in automatic speech recognition (ASR), supplementing existing multilingual ASR corpora , and advancing research in various problems such as low-resource ASR


Weakness of this paper

Several reviewers raised few concerns/limitations of this paper. By addressing these limitations, the paper could strengthen its experiment and expand impact.

- Scope of the dataset and generalizability: the dataset exclusively focuses on Indian languages, which provides rich diversity but limits its generalizability. Expanding the dataset with additional languages from resources like FLEURS or CommonVoice, especially those with minimal similarity to Indian languages, would improve its applicability and relevance. The paper has limited exploration in several important aspects, e.g.,  cross-lingual transfer or language relationships to have subgroup analyses based on geographic distance, language family, dialectal variations, vocabulary overlap, or phonetic variations, and investigating algorithmic adaptations or architecture-based approaches like Progressive Neural Networks (PNN), Piggyback (PB), and Learning to Prompt (L2P).
- Experiment settings could be improved: the task order is determined randomly, but analyzing the effects of task order and incorporating language similarity into task sequencing could add valuable insights; the dataset’s imbalance in data availability across languages may skew results and it's worthwhile to have some discussion about the effect.
- Concerns about novelty: the dataset’s design does not convincingly show the added value of Nirantar and its uniqueness for testing CL methods, as similar schemes could be applied to other datasets, such as phased releases of CommonVoice. Many of the works were also built upon existing efforts such as IndicVoices, Language-Incremental (LIL) and Domain-Incremental (DIL) setting, with some incremental changes.

**Additional Comments On Reviewer Discussion:**

In addition to above weaknesses, reviewers also raised some other weaknesses and suggested improvements (e.g., explanations and justifications for certain experimental choices, improving writing, additional statistics about language, domain, and episode , and pre-CL performance metrics) during rebuttal. Some of the weakness have been improved / somewhat addressed during rebuttal session (e.g., further explanation and justification on the questions raised by reviewers, more experiment results added). Although some review rating was raised, the rating averaged over all reviewers are still at borderline. I think the session is too short and some weaknesses are hard to address in such a short period of time. Also there is a general concern about the significance of novelty. Plus the main contribution of this paper is on the creation of dataset; which I believe better venue than ICLR exists for such efforts. Given the high bar of ICLR, I think the paper is still of limited interests to the audience , and thus I recommend to reject the paper, and the authors to re-work on these weakness and re-submitting to future conferences.

---

### Decision · Program_Chairs · 2025-01-22

Reject